# Computing Voting Rules with Improvement Feedback

**Evi Micha** [1]    **Vasilis Varsamis** [1]

## Abstract

Aggregating preferences under incomplete or constrained feedback is a fundamental problem in social choice and related domains. While prior work has established strong impossibility results for pairwise comparisons, this paper extends the inquiry to improvement feedback, where voters express incremental adjustments rather than complete preferences. We provide a complete characterization of the positional scoring rules that can be computed given improvement feedback. Interestingly, while plurality is learnable under improvement feedback—unlike with pairwise feedback—strong impossibility results persist for many other positional scoring rules. Furthermore, we show that improvement feedback, unlike pairwise feedback, does not suffice for the computation of any Condorcet-consistent rule. We complement our theoretical findings with experimental results, providing further insights into the practical implications of improvement feedback for preference aggregation.

## 1. Introduction

Classical social choice theory assumes that voters provide complete rankings of all candidates, which are then aggregated by a voting rule to select a winner (Brandt et al., 2016). However, this assumption becomes impractical in many real-world scenarios involving a large number of candidates, as voters may be unable or unwilling to rank all of them. For example, in deliberation platforms like Polis (Small et al., 2021), users provide pairwise comparisons over a limited subset of opinions rather than full rankings. Similarly, in Reinforcement Learning from Human Feedback (RLHF)—a methodology widely used to fine-tune large language models (LLMs)—feedback often takes the form of comparison queries between pairs of outputs (Ouyang et al., 2022; Christiano et al., 2017).

Despite their widespread use, pairwise or $t$-wise comparisons—rankings of $t$ candidates—face fundamental limitations. Recent work by Halpern et al. (2024) shows that, even under ideal conditions where the preferences of the population over every $t$-wise query are fully known (i.e., for every ranking of $t$ candidates, the proportion of the population that agrees with it is known), the winner under common voting rules cannot be determined. For example, the plurality winner cannot be reliably identified, and even randomized algorithms fail to perform better than random guessing.

Motivated by these limitations, we explore an alternative type of feedback known as *improvement feedback*. Unlike elicitation methods based on rankings or pairwise comparisons, improvement feedback enables agents to iteratively refine an initial suggestion. For instance, in RLHF applications, improvement feedback could involve a user modifying a draft generated by an LLM to better align with their preferences (Schick et al., 2023; Jin et al., 2023). Similarly, in robotics, users might iteratively adjust a robot's trajectory or behavior to achieve their desired outcome (Bajcsy et al., 2018; Yang et al., 2024).

This form of feedback aligns naturally with the framework of preference-based reinforcement learning (PbRL), which infers preferences from relative judgments rather than explicit reward signals. While PbRL typically relies on pairwise comparisons (e.g., preferring trajectory A over B), improvement feedback can be interpreted as the case where the preferred option is an improved version of the original. This connection is particularly salient in coactive learning frameworks (Shivaswamy & Joachims, 2015; Tucker et al., 2024), where users typically refine queried options through targeted adjustments rather than identifying the optimal candidate outright-—often because they may not even know what the optimal choice is.

A simple observation is that improvement feedback offers a promising pathway to address some challenges posed by $t$-wise comparisons. For example, unlike pairwise feedback—which struggles to compute the plurality winner—improvement feedback enables users to iteratively refine a suggested candidate. As users provide targeted adjustments, they gradually reveal their top preferences, ultimately

[1]Thomas Lord Department of Computer Science, University of Southern California, Los Angeles, California, USA. Correspondence to: Evi Micha <evi.micha@usc.edu>, Vasilis Varsamis <varsamis@usc.edu>.

*Proceedings of the $42^{nd}$ International Conference on Machine Learning*, Vancouver, Canada. PMLR 267, 2025. Copyright 2025 by the author(s).

enabling the identification of the plurality winner through this iterative process. This raises the following questions:

> Can other voting rules be computed using improvement feedback? For which voting rules is improvement feedback more effective than pairwise comparisons, and vice versa?

## 1.1. Our Contribution

We model improvement feedback as a process in which, when a user or agent is queried with a candidate ranked at position $i$ in their preference order or ranking, they return, with some probability, a candidate ranked at position $j$, where $j < i$. The likelihood of returning the candidate at position $j$ decreases as the distance $i - j$ increases, reflecting the agent's tendency to provide localized improvements. To formalize this, we introduce the concept of $t$-*improvement feedback*, where a user refines a queried option by selecting a better candidate from the $t$-*above neighborhood* of the queried candidate in their preference ranking—i.e., a candidate within the $t$ positions above the queried candidate. The $t$-*improvement feedback distribution* specifies the probabilities of selecting a candidate from this neighborhood. The parameter $t$ defines the size of this neighborhood, capturing how much better the returned candidate can be relative to the queried one. In practice, $t$ is typically much smaller than the total number of candidates, $m$, reflecting the limited cognitive and practical effort users are willing to expend.

Our goal is to investigate whether winners under various voting rules can be identified using $t$-*improvement feedback queries* over the underlying preferences of the agents, which are unknown to the algorithm. We consider an idealized setting, similar to (Halpern et al., 2024), where the algorithm has access to the full statistical distribution of feedback responses. Specifically, with a sufficiently large number of $t$-improvement feedback queries, the algorithm knows the exact probability of receiving candidate $b$ as feedback when querying candidate $a$, given the population's preferences and the underlying $t$-improvement feedback distribution. This idealized assumption strengthens our impossibility results and models scenarios where such statistical information is available or can be effectively estimated.

In Section 4, we study the well-known family of positional scoring rules and provide a complete characterization of the rules that are learnable under $t$-improvement feedback queries, for all practical values of $t$. We show that beyond plurality and a specific positional scoring rule uniquely determined by the $t$-improvement feedback distribution, as well as any linear combination of the two, no other positional scoring rule is learnable using $t$-improvement feedback queries for *any* value of $t \leq {}^m\!/_2 - 2$. In fact, we show that even randomized algorithms cannot reliably identify the correct winner with a probability greater than $1/m$

in this case. As discussed in the introduction, in practical settings $t \ll m$, thus making these findings particularly relevant. We also extend these negative results for every $t \in [m-1]$ under the uniform $t$-improvement distribution, where when a candidate $a$ is queried, a candidate from its $t$-above neighborhood is returned uniformly at random.

In Section 5, we turn our attention to Condorcet-consistent rules, which select the Condorcet winner whenever one exists. While pairwise comparisons suffice to identify the Condorcet winner, we surprisingly show that under $t$-improvement feedback, no (randomized) algorithm can determine the Condorcet winner with probability greater than $1/m$. This result applies under the uniform $t$-improvement feedback model for all values of $t$ and extends to all $t \leq {}^m\!/_2 - 2$ for almost every $t$-improvement feedback distribution. We also discuss the one specific exception to this result in detail.

These theoretical results indicate that while $t$-improvement feedback is particularly effective for identifying the plurality winner, it is insufficient for computing many other widely studied rules in social choice theory, such as Kemeny, Copeland, or Borda, which, by contrast, can be identified using pairwise comparison feedback.

Lastly, in Section 6, we compare the two types of feedback through simulations. Interestingly, contrary to the theoretical results, $t$-improvement feedback queries turn to be more efficient in some cases for implementing rules—such as Copeland or Borda—that are learnable from pairwise comparison feedback but are not implementable by $t$-improvement feedback in the theoretical worst case.

## 1.2. Related Work

Our work contributes to the growing body of research on decision-making with partial access to votes. Filmus & Oren (2014) and Oren et al. (2013) studied $t$-top queries, where each agent reports their top $t$-candidates, and investigated how large $t$ must be to reliably identify the correct winner under various voting rules. Similarly, Bentert & Skowron (2020) examined $t$-wise comparison queries, analyzing which voting rules can be implemented with this feedback. More recently, Halpern et al. (2024) provided a complete characterization of positional scoring rules that can be computed using $t$-wise comparison feedback.

These works build on foundational results regarding pairwise comparisons, which are often represented as (weighted) tournament graphs. Tournament graphs serve as the primary input for many well-known voting rules, including Borda count and several Condorcet-consistent rules, such as Copeland, Kemeny, and Minimax (Brandt et al., 2016). Our work builds on and extends the results of Halpern et al. (2024), which we discuss in greater detail later. To the best

of our knowledge, no prior work has studied improvement feedback queries for identifying the correct winner under different voting rules.

The query complexity of identifying the correct candidate has also been studied over different queries. For example, the query complexity of tournament graphs has been studied in the context of identifying Condorcet winners (Procaccia, 2008). Other works have explored the query complexity of complete rankings, focusing either on identifying the winner (Dey & Bhattacharyya, 2015) or recovering the full ranking (Micha & Shah, 2020) of different voting rules. In contrast, our work focuses on information-theoretic impossibilities, assuming perfect access to the feedback model under consideration, and does not analyze the sample complexity of learning voting rules.

Our work is also related to frameworks like coactive learning (Shivaswamy & Joachims, 2015) and interactive preference learning (Tucker et al., 2024), which focus on scenarios where users provide incremental improvements rather than global rankings. However, these frameworks aim to minimize regret and learn a good candidate for a single user through iterative interaction. In contrast, we investigate whether improvement feedback is sufficient to identify the correct candidate when preferences are heterogeneous across a population.

## 2. Model

For $k \in \mathbb{N}$, let $[k] = \{1, 2, \ldots, k\}$. We consider a set $M = \{a_1, \ldots, a_m\}$ of $m$ candidates. A ranking or preference $\sigma$ over the candidates is a bijection $\sigma : [m] \to M$, where $\sigma(i)$ returns the candidate that is the $i$-th most preferred candidate in $\sigma$ and $\sigma^{-1}(a)$ returns the position of candidate $a$ in $\sigma$. We denote by $\mathcal{L}(M)$ the set of all $m!$ possible rankings over $M$. We use $a \succ_\sigma b$ to denote that candidate $a$ is ranked above candidate $b$ under ranking $\sigma$.

Let $\Delta(\mathcal{L}(M))$ be the set of probability distributions over $\mathcal{L}(M)$. A preference profile corresponds to a distribution $D \in \Delta(\mathcal{L}(M))$ which represents the proportion of users in the population that have each ranking. For example, if $\Pr_{\sigma \sim D}[\sigma = a_1 \succ \ldots \succ a_m] = 1/3$ this means that $1/3$ of the population holds the ranking $a_1 \succ \ldots \succ a_m$.

Given a permutation over the candidates $\pi : M \to M$, we define $\pi \circ \sigma$ as the ranking obtained by permuting the candidates in $\sigma$ according to $\pi$. The special case $\pi_{ab}$ swaps only candidates $a$ and $b$, such that $\pi_{ab}(a) = b$, $\pi_{ab}(b) = a$, and $\pi_{ab}(c) = c$ for all $c \neq a, b$. For preference profiles, we denote by $\pi \circ D$ the preference profile induced by sampling $\sigma \sim D$ and outputting $\pi \circ \sigma$. Lastly, for swapping $a$ and $b$, $D^{a \leftrightarrow b} = \pi_{ab} \circ D$.

**Voting Rules.** A voting rule is a function $f : \Delta(\mathcal{L}(M)) \to M$ that takes as input a preference profile and outputs a candidate as the winner. We call the winner of $f$ the $f$-winner. In this work, we focus on two main families of voting rules.

The first family is the family of *positional scoring rules*, which are defined using a scoring vector $\vec{s} = (s_1, s_2, \ldots, s_m) \in \mathbb{R}^m$ with $s_i \geq s_{i+1}$ for all $i \in [m-1]$ and $s_1 > s_m$. Without loss of generality, we usually assume $s_m = 0$. The score of a candidate $a \in M$ under a positional scoring rule $f_{\vec{s}}$, parametrized by a scoring vector $\vec{s}$, given a preference profile $D \in \Delta(\mathcal{L}(M))$, is defined as follows:

$$sc_{\vec{s}}(a, D) = \sum_{i=1}^{m} \Pr_{\sigma \sim D}[\sigma^{-1}(a) = i] \cdot s_i.$$

The rule $f_{\vec{s}}$ then returns the candidate with the highest score as the winner, breaking ties arbitrarily. Some well-known positional scoring rules are plurality, parametrized by $\vec{s}_{plu} = (1, 0, \ldots, 0)$; veto, parametrized by $\vec{s}_{veto} = (1, \ldots, 1, 0)$; and Borda count, parametrized by $\vec{s}_{Borda} = (m-1, m-2, \ldots, 0)$.

The second family is the family of *Condorcet-consistent rules*. A candidate $a \in M$ is called the *Condorcet winner* if they beat every other candidate in pairwise majority comparisons, i.e., $\Pr_{\sigma \sim D}[a \succ_\sigma b] > 0.5$ for all $b \in M \setminus \{a\}$. A Condorcet-consistent voting rule $f$ selects the Condorcet winner whenever one exists. Famous examples of Condorcet-consistent rules include Copeland's Rule, Kemeny's Rule, the Minimax Rule, Ranked pairs, and many others (Brandt et al., 2016).

**Improvement Feedback Distribution.** In the improvement feedback setting, when an agent is queried with a candidate $a$ ranked at position $i$ in the agent's preference ranking $\sigma$, the agent returns an improved candidate $b$ ranked at position $j$, where $j < i$. The candidates ranked in positions $j \in [\max(i-t, 1), i-1]$ are referred to as the *t-above neighborhood* of $\sigma(i)$, representing the set of candidates that are strictly preferred but within $t$ positions above the queried candidate. The only exception occurs when $a$ is the agent's first choice (i.e., $i = 1$), in which case no improvement is provided, and the agent returns $a$.

More formally, for a fixed parameter $t \leq m - 1$, the *t-improvement feedback distribution* defines the probability of returning a candidate $\sigma(j)$ when querying $\sigma(i)$, denoted as $p_{i,j}^t$. The distribution satisfies the following properties:

- $p_{i,j}^t > 0$ only if $j < i$, ensuring that the returned candidates are strictly preferred to the queried candidate. The only exception is $p_{1,1}^t = 1$, reflecting that no improvement is possible when querying the top-ranked

candidate.

- The probabilities sum to 1 over the candidates in the *t-above neighborhood*:

$$\sum_{j=\max(i-t,1)}^{i-1} p_{i,j}^t = 1, \quad \forall i > 1.$$

We pay special attention to the *uniform t-improvement feedback distribution*, where the agent selects an improved candidate uniformly at random from the $t$-above neighborhood:

$$p_{i,j}^t = \begin{cases} \frac{1}{\min(t,i-1)}, & \text{if } 0 < i - j \le t, \\ 0, & \text{otherwise.} \end{cases}$$

We also denote $P_i^t$ as the cumulative probability that a candidate ranked at position $i$ is returned as an improvement when querying a candidate ranked at any position below $i$, i.e.,

$$P_i^t = \sum_{j=i+1}^{\min(m,i+t)} p_{j,i}^t.$$

Intuitively, $P_i^t$ represents the overall likelihood that a candidate at position $i$ is selected through the improvement feedback process, summing over the probabilities of being reached from any candidate ranked below it. Notably, $P_m^t = 0$ since there is no candidate ranked below the last position $m$ that could return it as an improvement.

**Improvement Feedback Queries.** We consider algorithms that make $t$-improvement feedback queries over the preference profile $D$. For $x \in M$ and $\sigma \in \mathcal{L}(M)$, we denote by $q_\sigma^t(x)$ the candidate returned when querying $x$ in $\sigma$ under $t$-improvement feedback. Specifically, we assume that an algorithm has access to the exact probability of observing $a \in M$ when querying $b \in M$, denoted by $\Pr_{\sigma \sim D}[q_\sigma^t(b) = a]$, and given by:

$$\Pr_{\sigma \sim D}[q_\sigma^t(b) = a] =$$
$$\sum_{i=1}^{m-1} \sum_{j=i+1}^{\min(m,i+t)} p_{j,i}^t \cdot \Pr_{\sigma \sim D}[\sigma^{-1}(a) = i, \sigma^{-1}(b) = j].$$

This expression considers all possible positions of $a$ and $b$ in the ranking, weighting the feedback probability $p_{j,i}^t$ by the likelihood that $a$ and $b$ occupy these positions under the population distribution $D$. Additionally, we denote by $\Pr_{\sigma \sim D}[q^t(a) = a]$ the probability of not returning any improvement, as $a$ is already ranked first. This probability reflects the likelihood that $a$ is the top-ranked candidate in

the preference profile. Therefore, as discussed in the introduction, when this information is available, the plurality winner can be directly determined.

Having direct access to this distribution provides strictly more information than an approximation based on random or adjusted queries (where, upon querying $a$, the algorithm observes $b$ with some probability). Since our impossibility results hold even under this idealized setting, they provide fundamental limits that also apply to real-world scenarios with finite queries.

$t$**-Indistinguishable Preference Profiles.** Two preference profiles $D_1, D_2 \in \Delta(\mathcal{L}(M))$ are said to be $t$-indistinguishable if, for every pair of candidates $a, b \in M$, the probability of receiving $a$ as feedback when querying $b$ is the same under both profiles. Formally,

$$\Pr_{\sigma \sim D_1}[q_\sigma^t(b) = a] = \Pr_{\sigma \sim D_2}[q_\sigma^t(b) = a].$$

Thus, no algorithm with access only to $t$-improvement feedback can distinguish between $D_1$ and $D_2$.

In our results, we will focus on the $t$-indistinguishability between a preference profile $D$ and its swapped version $D^{a \leftrightarrow b}$. In this case, when the feedback probabilities satisfy

$$\Pr_{\sigma \sim D}[q_\sigma^t(x) = y] = \Pr_{\sigma \sim D^{a \leftrightarrow b}}[q_\sigma^t(x) = y]$$
$$= \Pr_{\sigma \sim D}[q_\sigma^t(\pi_{ab}(x)) = \pi_{ab}(y)],$$

for all $x, y \in M$, then the profiles are $t$-indistinguishable.

## 3. Main $t-$Indistinguishable Preference Profiles

We begin by constructing a family of preference profiles, denoted $D_{i,j,\ell}$, which will serve as the foundation for our impossibility results.

**Definition 3.1.** Let $D_{i,j,\ell}$ be a preference profile defined with respect to two candidates $a$ and $b$, as follows:

1. With probability $p$, candidate $a$ is fixed at position $i$ (where $i > 1$), and candidate $b$ is fixed at position $j$ (where $j - i > t$).

2. With probability $1 - p$, candidate $a$ is fixed at position $m$, and candidate $b$ is fixed at position $\ell$, where $1 < \ell < m - t$.

3. Select a uniformly random ranking $\tau^{S-ab}$ of all remaining candidates (excluding $a$ and $b$). This ranking determines their relative order, and they are then assigned to the unoccupied positions.

The proof of the following lemma can be found in Appendix A

**Lemma 3.2.** *For any $m \geq 4$ and any $t \in [m-1]$, the preference profiles $D_{i,j,\ell}$ and $D_{i,j,\ell}^{a \leftrightarrow b}$ are $t$-indistinguishable when $p = \frac{P_\ell^t}{P_i^t - P_j^t + P_\ell^t}$.*

We will also rely on the following technical lemma to establish our impossibility results. Intuitively, this lemma suggests that if there exist $m$ distinct preference profiles that are $t$-indistinguishable from one another, and each has a different $f$-winner under some voting rule $f$, then no randomized algorithm can identify the correct winner with a probability greater than $1/m$. This result corresponds to Lemma 4.2 of (Halpern et al., 2024), but we restate and prove it here for completeness.

**Lemma 3.3.** *Fix any voting rule $f$ and suppose there is a family of $m$ preference profiles that are all $t$-indistinguishable from one another, but each has a distinct singleton $f$-winner. Then, for all (possibly randomized) algorithms $A$ that have access to $t$-improvement feedback, there is a preference profile with a unique $f$-winning candidate $a$, such that $A$ outputs $a$ with probability at most $1/m$.*

*Proof.* We denote with $\{D^c\}_{c \in M}$ the set of $m$ preference profiles that are all $t$-indistinguishable and $c$ is the unique winner in $D^c$ under $f$. When the algorithm is applied to any of these preference profiles, it will receive the exact same feedback, and therefore the output should be the same across them. Due to the pigeonhole principle, there must be a candidate $a$ that is returned as the winner with probability at most $1/m$, and hence $D^a$ satisfies the desired property. $\square$

## 4. Positional Scoring Rules

In this section, we consider the family of positional scoring rules and provide a complete characterization of the rules that are learnable under $t$-improvement feedback queries. For general $t$-improvement feedback distributions, this characterization applies for all $t \leq m/2 - 2$, while for the uniform $t$-improvement feedback distribution, it extends to all $t \in [m-1]$.

In Section 4.1, we show that for *any* value of $t \leq m/2 - 2$, the only learnable scoring vectors are linear combinations of $\vec{s}_t^*$ and $\vec{s}_{\text{plu}}$, where $\vec{s}_t^* = (P_1^t, \ldots, P_m^t)$. In fact, we prove that even randomized algorithms cannot identify the correct winner for any scoring vector $\vec{s}$ outside the $\text{span}(\vec{s}_t^*, \vec{s}_{\text{plu}})$ with a probability greater than $1/m$. In Section 4.2, we extend these negative results to cover all values of $t$ under the uniform $t$-improvement feedback distribution.

For showing our negative results, we start with the following lemma, which establishes the conditions under which a family of $t$-indistinguishable profiles, as described in Lemma 3.3, can be constructed. This lemma closely resembles Lemma 4.1 of Halpern et al. (2024) and its proof can be found in Appendix B

**Lemma 4.1.** *Fix a scoring vector $\vec{s}$. Let $D$ and $D^{a \leftrightarrow b}$ be two $t$-indistinguishable preference profiles and let $a, b \in M$ be two candidates such that $sc_{\vec{s}}(a, D) \neq sc_{\vec{s}}(b, D)$. Then, there exists a family of preference profiles $\{D^c\}_{c \in M}$ such that all profiles are $t$-indistinguishable from one another, but each candidate $c \in M$ uniquely maximizes $sc_{\vec{s}}(c, D^c)$.*

### 4.1. General $t-$Improvement Feedback Distribution

Here, we consider general $t$-improvement feedback distributions. To provide a complete characterization of the scoring rules that are learnable under $t$-improvement feedback queries, we first prove the following lemma.

**Lemma 4.2.** *For any $m \geq 6$, any $t \leq m/2 - 2$, any pair of candidates $a, b \in M$ and any scoring vector $\vec{s} \notin \text{span}(\vec{s}_{plu}, \vec{s}_t^*)$, there exists a preference profile $D$ such that $sc_{\vec{s}}(a, D) \neq sc_{\vec{s}}(b, D)$, and $D$ and $D^{a \leftrightarrow b}$ are $t$-indistinguishable.*

*Proof.* Consider the preference profile $D_{i,m,i+1}$ as defined in Definition 3.1. From Lemma 3.2, we have that for all $i \in \{2, \ldots, m - t - 2\}$, $D_{i,m,i+1}$ and $D_{i,m,i+1}^{a \leftrightarrow b}$ are $t$-indistinguishable for every $t \leq m - 4$, when

$$p = \frac{P_{i+1}^t}{P_i^t - P_m^t + P_{i+1}^t} = \frac{P_{i+1}^t}{P_i^t + P_{i+1}^t}.$$

If it holds that $sc_{\vec{s}}(a, D_{i,m,i+1}) \neq sc_{\vec{s}}(b, D_{i,m,i+1})$ for some $i \in \{2, \ldots, m - t - 2\}$, then setting $D = D_{i,m,i+1}$ satisfies the lemma.

Now, suppose that for every $i \in \{2, \ldots, m - t - 2\}$, we have

$$sc_{\vec{s}}(a, D_{i,m,i+1}) = sc_{\vec{s}}(b, D_{i,m,i+1}).$$

This implies that

$$p \cdot s_i = (1 - p) \cdot s_{i+1} \implies p = \frac{s_{i+1}}{s_i + s_{i+1}}.$$

Thus, we have

$$p = \frac{P_{i+1}^t}{P_i^t + P_{i+1}^t} = \frac{s_{i+1}}{s_i + s_{i+1}} \implies \frac{s_{i+1}}{P_{i+1}^t} = \frac{s_i}{P_i^t} = \lambda, \tag{1}$$

for all $i \in \{2, \ldots, m - t - 2\}$.

Assuming $s_i/P_i^t = \lambda$ for all $i \in \{2, \ldots, m - t - 1\}$, we next consider the preference profile $D_{2,i,2}$ as defined in Definition 3.1. From Lemma 3.2, we have that for

all $i \in \{m - t, \ldots, m - 1\}$, $D_{2,i,2}$ and $D_{2,i,2}^{a \leftrightarrow b}$ are $t$-indistinguishable for every $t \leq m/2 - 2$, when

$$p = \frac{P_2^t}{2P_2^t - P_i^t}.$$

If it holds that $sc_{\vec{s}}(a, D_{2,i,2}) \neq sc_{\vec{s}}(b, D_{2,i,2})$ for some $i \in \{m - t, \ldots, m - 1\}$, then setting $D = D_{2,i,2}$ satisfies the lemma.

Now, suppose that for every $i \in \{m - t, \ldots, m - 1\}$, we have

$$sc_{\vec{s}}(a, D_{2,i,2}) = sc_{\vec{s}}(b, D_{2,i,2}).$$

This implies

$$p \cdot s_2 = (1 - p) \cdot s_2 + p \cdot s_i \implies p = \frac{s_2}{2s_2 - s_i}.$$

Thus, we obtain

$$p = \frac{P_2^t}{2P_2^t - P_i^t} = \frac{s_2}{2s_2 - s_i}.$$

From Equation (1), we know that $\frac{s_2}{P_2^t} = \lambda$. Then by combining Equation (1) and the above equality, we get that $\frac{s_i}{P_i^t} = \lambda$ for all $i \in \{2, \ldots, m - 1\}$. This means that, unless $\frac{s_i}{P_i^t} = \lambda$ for all $i \in \{2, \ldots, m - 1\}$, which implies that $\vec{s} \in \mathrm{span}(\vec{s}_{\mathrm{plu}}, \vec{s}_t^*)$, there exists a preference profile $D$ satisfying the conditions of the lemma and the lemma follows. $\qquad\square$

Next, we show that if $\vec{s} \in span(\vec{s}_t^*, \vec{s}_{plu})$, then the scores of all the candidates under $\vec{s}$ can be computed with access to $t$-improvement feedback queries. The proof of the following lemma can be found in Appendix C

**Lemma 4.3.** *For any $t \in [m - 1]$ and any $\vec{s} \in span(\vec{s}_{plu}, \vec{s}_t^*)$, given $a \in M$ and $D \in \Delta(\mathcal{L}(M))$, it is possible to calculate $sc_{\vec{s}}(a, D)$ with $t$-improvement feedback queries.*

Now, we are ready to prove the following theorem.

**Theorem 4.4.** *For any $m \geq 6$, any $t \leq m/2 - 2$, and any scoring vector $\vec{s}$:*

1. *If $\vec{s} \in \mathrm{span}(\vec{s}_t^*, \vec{s}_{plu})$, then using $t$-improvement feedback queries, the candidate that maximizes the score under $\vec{s}$ for any input profile $D$ can be found.*

2. *If $\vec{s} \notin \mathrm{span}(\vec{s}_t^*, \vec{s}_{plu})$, then no randomized algorithm with access to $t$-improvement feedback queries can output the candidate with the maximum score under $\vec{s}$ for any input profile $D$ with probability greater than $1/m$.*

*Proof.* The first part of the lemma immediately follows from Lemma 4.3.

For the second part, since in Lemma 4.2 we show the existence of a preference profile $D$ that is indistinguishable from $D^{a \leftrightarrow b}$ and has the desired properties, we can apply Lemma 4.1 for constructing a family of preference profiles that are $t$-indistinguishable and each of them has a different candidate as a winner under a scoring rule not in $span(\vec{s}_t^*, \vec{s}_{plu})$. Then, by applying Lemma 3.3, the theorem follows. $\qquad\square$

An interesting observation arises when $t = 1$. In this case, $P_i^t = 1$ for all $i \in [m - 1]$, since each candidate ranked at position $i$ can only be returned as an improvement of a candidate ranked at position $i + 1$, and when a candidate at position $i + 1$ is queried, the agent always returns the candidate at position $i$ with probability 1. As a result, the scoring vector $\vec{s}_t^* = (1, \ldots, 1, 0)$, which corresponds to the veto scoring rule. Therefore, when $t = 1$, we can learn all rules within the span of plurality and veto. However, for larger values of $t$, the veto scoring rule is no longer learnable under $t$-improvement feedback queries.

### 4.2. Uniform $t-$Improvement Feedback Distribution

Here, we turn our attention to the uniform $t$-improvement feedback distribution. In this setting, we extend Lemma 4.2 to cover all values of $t$. To derive this result, we introduce an additional set of preference profiles, distinct from those in Definition 3.1, and carefully analyze the resulting cases. The proof of the following lemma can be found in Appendix D

**Lemma 4.5.** *Under uniform $t$-improvement feedback distribution, for any $t \in [m-1]$, any pair of candidates $a, b \in M$ and any scoring vector $\vec{s} \notin \mathrm{span}(\vec{s}_{plu}, \vec{s}_t^*)$, there exists a preference profile $D$ such that $sc_{\vec{s}}(a, D) \neq sc_{\vec{s}}(b, D)$, and $D$ and $D^{a \leftrightarrow b}$ are $t$-indistinguishable.*

From the above lemma, we immediately derive the following theorem in a manner similar to Theorem 4.4.

**Theorem 4.6.** *Under uniform $t$-improvement feedback distribution, for any $m \geq 6$, any $t \in [m - 1]$, and any scoring vector $\vec{s}$:*

1. *If $\vec{s} \in \mathrm{span}(\vec{s}_t^*, \vec{s}_{plu})$, then using $t$-improvement feedback queries, the candidate that maximizes the score under $\vec{s}$ for any input profile $D$ can be found.*

2. *If $\vec{s} \notin \mathrm{span}(\vec{s}_t^*, \vec{s}_{plu})$, then no randomized algorithm with access to $t$-improvement feedback queries can output the candidate with the maximum score under $\vec{s}$ for any input profile $D$ with probability greater than $1/m$.*

# 5. Condorcet Consistent Rules

In the previous section, we showed that $t$-improvement feedback queries do not help us overcome the impossibility results associated with positional scoring rules and pairwise comparison queries, except for plurality and a specific positional scoring rule determined by the $t$-improvement feedback distribution. In this section, we extend these negative results to the family of Condorcet-consistent rules.

In Section 5.1, we demonstrate that for every $t \in [m-1]$, no algorithm can reliably identify the Condorcet winner using uniform $t$-improvement feedback queries. In fact, even randomized algorithms cannot identify the correct candidate with a probability greater than $1/m$. In Section 5.2, we extend this negative result to *any* $t$-improvement feedback distributions for $t \leq m/2 - 2$, with only one exception, which we discuss in detail. These results contrast sharply with pairwise comparison queries, which are sufficient for identifying the Condorcet winner whenever one exists.

As in the case of positional scoring rules, to establish these negative results, we start with the following lemma, which establishes the conditions under which a set of $m$ $t$-indistinguishable profiles, as described in Lemma 3.3, can be constructed. Again, this lemma builds upon Lemma 4.1 of Halpern et al. (2024), employing the same method for constructing the set of distinct preference profiles. However, the condition that the initial preference profile must satisfy and the arguments used in the proof are significantly different. The proof of the following lemma can be found in Appendix E

**Lemma 5.1.** *Suppose there exist $a, b \in M$ and a preference profile $D$ such that $D$ and $D^{a \leftrightarrow b}$ are $t$-indistinguishable, and it holds that*

$$\sum_{x \in M \setminus \{a\}} \Pr_{\sigma \sim D} [a \succ_\sigma x] \neq \sum_{x \in M \setminus \{b\}} \Pr_{\sigma \sim D} [b \succ_\sigma x].$$

*Then, there exists a family of preference profiles $\{D^c\}_{c \in M}$ such that all preference profiles in the family are $t$-indistinguishable from one another, and each preference profile $D^c$ has $c$ as the Condorcet winner.*

## 5.1. Uniform $t-$Improvement Feedback Distribution

In the following lemma, we claim that there is a preference profile that satisfies the conditions of Lemma 5.1, when the algorithm has access to uniform $t$-improvement feedback queries, for $t \in [m-1]$. The proof is deferred to Appendix F.

**Lemma 5.2.** *For any $m \geq 13$ and any $t \in [m-1]$, and pair of candidates $a, b \in M$, there is a preference profile $D$*

*such that*

$$\sum_{x \in M \setminus \{a\}} \Pr_{\sigma \sim D} [a \succ_\sigma x] \neq \sum_{x \in M \setminus \{b\}} \Pr_{\sigma \sim D} [b \succ_\sigma x],$$

*and $D$ and $D^{a \leftrightarrow b}$ are $t$-indistinguishable under the uniform $t$-improvement feedback distribution.*

Since in Lemma 5.2 we show the existence of a preference profile $D$ that is indistinguishable from $D^{a \leftrightarrow b}$ and has the desired properties, we can apply Lemma 5.1 for constructing a family of preference profiles that are $t$-indistinguishable from one another and each of them has a different candidate as the Condorcet winner. Then, by applying Lemma 3.3 we get the following theorem.

**Theorem 5.3.** *For $m \geq 13$ and any $t \in [m-1]$, no randomized algorithm $A$ that has access to uniform $t$-improvement feedback queries outputs the unique Condorcet winner with probability more than $1/m$.*

## 5.2. General $t-$Improvement Feedback Distribution

Now, we extend the negative results to *every* $t$-improvement feedback distribution when $t \leq m/2 - 2$, with the exception of the special case where $P_i^t/P_{i+1}^t = (m-i)/(m-i-1)$ for all $i \in \{2, \ldots, m-1\}$. In Appendix H, we discuss in detail why this result cannot be applied in this case. We also show that for $t \leq m - 4$, no deterministic rule can identify the Condorcet winner under *any* $t$-improvement feedback queries. Thus, while the negative result does not extend to randomized algorithms for this specific $t$-improvement feedback distribution, it still holds for deterministic algorithms across all distributions and all values of $t \leq m - 4$.

To show the desired result, we use the following lemma, which is proved in Appendix G.

**Lemma 5.4.** *Unless $P_i^t/P_{i+1}^t = (m-i)/(m-i-1)$ for all $i \in \{2, \ldots, m-2\}$, then for any $m \geq 6$ and $t \leq m/2 - 2$, and pair of candidates $a, b \in M$, there is a preference profile $D$ such that*

$$\sum_{x \in M \setminus \{a\}} \Pr_{\sigma \sim D} [a \succ_\sigma x] \neq \sum_{x \in M \setminus \{b\}} \Pr_{\sigma \sim D} [b \succ_\sigma x],$$

*and $D$ and $D^{a \leftrightarrow b}$ are $t$-indistinguishable.*

As before, by combining Lemma 5.4, Lemma 5.1 and Lemma 3.3, we get the following theorem.

**Theorem 5.5.** *For $m \geq 6$ and $t \leq m/2 - 2$, unless the $t$-improvement feedback distribution satisfies $P_i^t/P_{i+1}^t = (m-i)/(m-i-1)$ for all $i \in \{2, \ldots, m-1\}$, no randomized algorithm $A$ with access to $t$-improvement feedback queries can identify the unique Condorcet winner with a probability exceeding $1/m$.*

# 6. Experiments

In the previous sections, we show that improvement feedback queries are not sufficient in the worst case to identify the winner under most positional scoring rules and any Condorcet-consistent rule. In this section, we empirically evaluate the effectiveness of $t$-improvement feedback queries compared to pairwise comparison queries over one positional scoring rule, namely Borda count, and one Condorcet consistent rule, namely Copeland's rule. Copeland's rule calculates the number of candidates each candidate defeats in pairwise comparisons and returns the candidate with the most wins. For both rules, access to the *weighted majority graph* suffices to determine the winner. In this graph, each candidate is represented as a node, and a directed edge $(a, b)$ is weighted by the *margin* by which candidate $a$ defeats candidate $b$ in a head-to-head comparison.

We evaluate the performance of the feedback mechanisms under three different distributions of rankings. The first is the impartial culture (IC), where every ranking in $\mathcal{L}(M)$ is equally likely to appear, providing a uniform distribution over rankings. The second distribution is derived from the Mallows model, which is parameterized by a central ranking $\sigma^*$ and a noise parameter $\phi \in [0, 1]$. The probability of a ranking $\sigma$ is proportional to $\phi^{d(\sigma, \sigma^*)}$, where $d(\sigma, \sigma^*)$ is the Kendall tau distance between $\sigma$ and $\sigma^*$. As $\phi$ approaches 0, the distribution concentrates most of the probability mass on $\sigma^*$, while as $\phi$ approaches 1, the distribution converges to the impartial culture. In our experiments, we set $\phi = 1/3$.

The third distribution is the Plackett-Luce (PL) model, which generates rankings through a sequential selection process where each candidate $a \in M$ is assigned a utility $u_a > 0$. The probability of selecting a candidate at any step is proportional to its utility. For our experiments, we assign utilities uniformly at random from the interval $[0, 1]$.

We consider three different $t$-improvement feedback distributions: (1) *Uniform distribution*: each candidate in the $t$-above neighborhood of the queried candidate is returned with equal probability, (2) *Linear decay distribution*: the probability of returning a candidate in the $t$-above neighborhood decreases linearly with the distance from the queried candidate and (3) *Exponential decay distribution*: the probability of returning a candidate in the $t$-above neighborhood decreases exponentially with the distance from the queried candidate.

We construct an estimate of the weighted majority graph using the two types of feedback by making one random query per user. Specifically, for the $t$-improvement feedback queries, upon randomly querying a candidate $a$, if the agent returns an alternative $b$ from $a$'s $t$-above neighborhood, we update our pairwise comparison estimates by adding the pair $(b, a)$. If the agent returns the queried candidate $a$ (indi-cating that it is their top choice), we update our estimates by adding all pairs $(a, c)$ for each $c \in M \setminus \{a\}$. For pairwise comparison queries, we randomly sample a pair of candidates $(a, b)$ and ask the agent to compare them, directly updating based on the comparison outcome.

Using the estimated weighted majority graph, we compute the Borda and Copeland winners—both derived from the same graph. We then evaluate the true score of each selected winner under the full preference profile and divide it by the true score of the actual winner, i.e., the candidate who would have been selected using complete information. This approximation ratio is averaged over 500 trials, and we report both the mean and standard deviation.

For all the experiments, we set the number of candidates $m = 20$ and vary the number of agents from 50 to 1000 in increments of 50. Here, we illustrate the results for the uniform improvement feedback distribution, and in Appendix I, we surprisingly show that all three improvement feedback distributions behave identically. We also set $t = 5$, and in Appendix I, we show that the results are quantitatively similar for different values of $t$. For each experiment, we iterate over 500 iterations[1].

In Figure 1, we plot the average approximation ratio along with standard deviations. We notice that under the IC model, the two types of feedback—$t$-improvement feedback queries and pairwise comparison queries—behave identically. We believe this is because rankings in IC are sampled uniformly at random, meaning there is no underlying structure or consistent patterns for either feedback mechanism to exploit. On the other hand, under the PL model, pairwise comparison queries provide a better approximation to the optimal score than $t$-improvement feedback queries, while under the Mallows model, the opposite is true. A positive explanation for these phenomena lies in the differences in how the models generate rankings. In the Mallows model, as the parameter $\phi$ decreases, the rankings converge more closely to the underlying ranking, making it more likely that the Borda and Copeland winners coincide with the plurality winner. Since $t$-improvement feedback is highly effective at identifying the top alternative—each time we query candidate $a$ and receive $a$ as a response (indicating it is the top choice) we directly increase $a$'s score relative to all other candidates—it excels in this setting. On the other hand, under the PL model, utilities for candidates are drawn uniformly at random, which implies that the relative probabilities of candidates being ranked higher or lower can exhibit more variance compared to the Mallows model, especially when the PL parameters are widely spread out. As a result, the probability that two candidates are frequently ranked close

---

[1]The code for the experimental part can be found in https://github.com/VasilisVar00/Computing-Voting-Rules-with-Improvement-Feedback

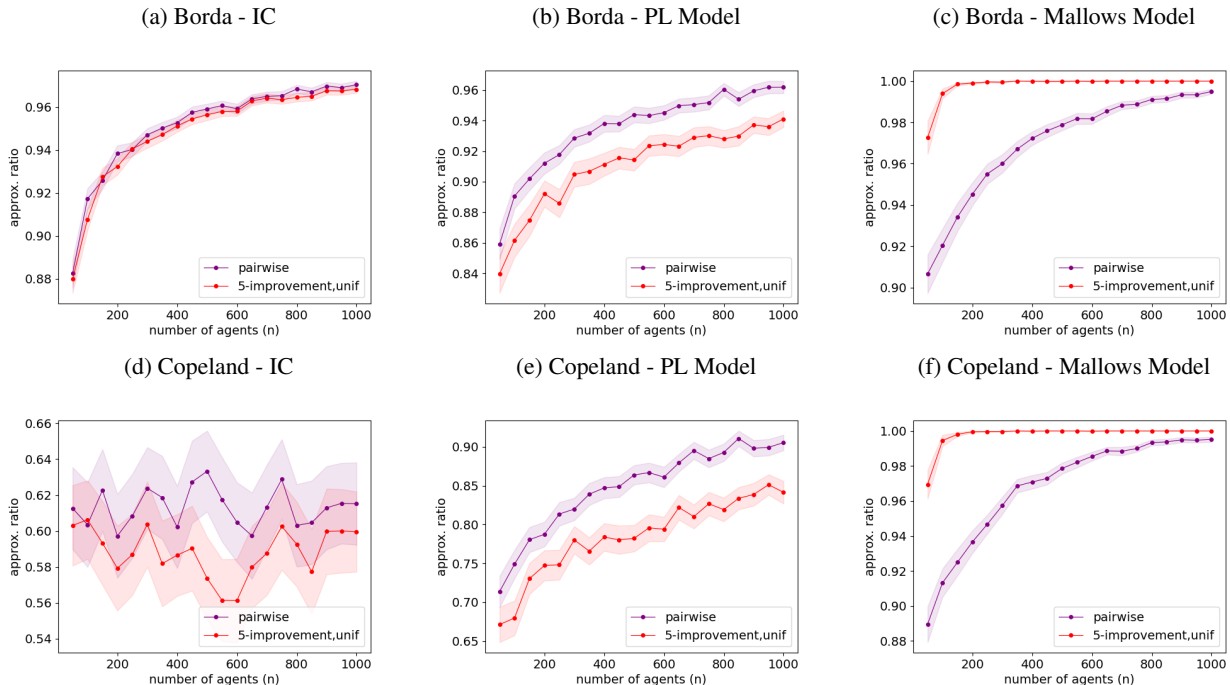

Figure 1: Approximation ratio of the optimal Borda score (top) and the optimal Copeland score (bottom) for different numbers of agents and various underlying ranking distributions, with $t = 5$ under the uniform improvement feedback distribution.

to each other (or even swapped in rank) is higher under the PL model. This increased variance in relative rankings benefits pairwise comparison queries because they directly compare the probabilities of individual pairs and can more effectively distinguish small differences in candidates' utilities. In summary, it seems that the effectiveness of each type of feedback is inherently tied to the structure of the underlying preference profile.

## 7. Discussion

Our work examines fundamental limitations in inferring collective decisions from restricted forms of feedback, a setting that frequently arises in real-world applications where full preference information is costly or impractical to elicit. Understanding which social choice rules exhibit robustness under such informational constraints is critical for the principled design of aggregation mechanisms that are both implementable and representative.

An immediate question that is left open is whether our negative results extend to all values of $t$ under general $t$-improvement feedback distributions. Additionally, it would be interesting to explore how the results change when agents are characterized by different $t$ values, reflecting varying levels of effort in providing feedback. Another promising direction involves hybrid feedback mechanisms that com-

bine $t$-improvement feedback with other methods, such as pairwise comparisons or partial rankings, potentially overcoming the limitations highlighted in our work by leveraging the complementary strengths of different feedback types.

While our negative results hold under the assumption that voters report their preferences truthfully and introducing strategic behavior would weaken these impossibility results, understanding strategic behavior in this setting is an interesting future direction as well.

## Impact Statement

This paper aims to advance the theoretical foundations of decision-making through iterative feedback mechanisms, with potential applications in Machine Learning, AI alignment, and democratic decision-making systems. Our findings are primarily theoretical, focusing on understanding the limitations and capabilities of feedback-based preference aggregation under various voting rules. As such, the immediate societal consequences of this work are limited to contexts where human feedback is used to guide large-scale automated systems.

One potential societal risk tied to our work stems from the misapplication of the theoretical results. The core of our paper demonstrates that certain feedback mechanisms fail to reliably learn voting outcomes in the *worst case*, even when

provided with ideal data access. If misunderstood or improperly communicated, these findings could inadvertently justify the dismissal of valuable feedback mechanisms or discourage the adoption of participatory decision-making systems in settings where iterative input from users could still provide meaningful outcomes.

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

# A. Proof of Lemma 3.2

*Proof.* Let $a, b$ be the candidates of the statement.

Since $j - i > t$ and $\ell < m - t$, $a, b$ are more than $t$ positions apart in both rankings. Therefore, it holds that $Pr_{\sigma \sim D_{i,j,\ell}}[q_\sigma^t(a) = b] = Pr_{\sigma \sim D_{i,j,\ell}}[q_\sigma^t(b) = a] = 0$. Moreover, since $i > 1$ and $\ell > 1$, it holds that $Pr_{\sigma \sim D_{i,j,\ell}}[q_\sigma^t(a) = a] = Pr_{\sigma \sim D_{i,j,\ell}}[q_\sigma^t(b) = b] = 0$.

Furthermore, for every $x \in S_{-ab}$, the following holds:

$$
\begin{aligned}
Pr_{\sigma \sim D_{i,j,\ell}}[q_\sigma^t(a) = x] = p \cdot & \sum_{k=\max(i-t,1)}^{i-1} Pr[\tau^{S_{-ab}}(k) = x] \cdot Pr_{\sigma \sim D_{i,j,\ell}}[q_\sigma^t(a) = x \mid \tau^{S_{-ab}}(k) = x] \\
+ (1-p) \cdot & \sum_{k=m-t-1}^{m-2} Pr[\tau^{S_{-ab}}(k) = x] \cdot Pr_{\sigma \sim D_{i,j,\ell}}[q_\sigma^t(a) = x \mid \tau^{S_{-ab}}(k) = x] \\
= p \cdot \frac{1}{m-2} \cdot & \sum_{k=\max(i-t,1)}^{i-1} Pr_{\sigma \sim D_{i,j,\ell}}[q_\sigma^t(a) = x \mid \tau^{S_{-ab}}(k) = x] \\
+ (1-p) \cdot \frac{1}{m-2} \cdot & \sum_{k=m-t-1}^{m-2} Pr_{\sigma \sim D_{i,j,\ell}}[q_\sigma^t(a) = x \mid \tau^{S_{-ab}}(k) = x] \\
= p \cdot \frac{1}{m-2} \cdot & \sum_{k=\max(i-t,1)}^{i-1} p_{i,k}^t \\
+ (1-p) \cdot \frac{1}{m-2} \cdot & \sum_{k=m-t-1}^{m-2} p_{m,k+1}^t \\
= p \cdot \frac{1}{m-2} + (1-p) \cdot & \frac{1}{m-2} = \frac{1}{m-2},
\end{aligned}
$$

where the second equality holds because $\tau^{S_{-ab}}$ is chosen uniformly at random, and the second last equality holds because the sum of probabilities of returning a candidate ranked above the queried candidate, over all valid positions within the $t$-above neighborhood, equals 1.

Similarly,

$$Pr_{\sigma \sim D_{i,j,\ell}}[q_\sigma^t(b) = x] = p \cdot \sum_{k=\max(j-t,1)-1}^{j-2} Pr[\tau^{S-ab}(k) = x] \cdot Pr_{\sigma \sim D_{i,j,\ell}}[q_\sigma^t(b) = x \mid \tau^{S-ab}(k) = x]$$

$$+ (1-p) \cdot \sum_{k=\max(\ell-t,1)}^{\ell-1} Pr[\tau^{S-ab}(k) = x] \cdot Pr_{\sigma \sim D_{i,j,\ell}}[q_\sigma^t(b) = x \mid \tau^{S-ab}(k) = x]$$

$$= p \cdot \frac{1}{m-2} \cdot \sum_{k=\max(j-t,1)-1}^{j-2} Pr_{\sigma \sim D_{i,j,\ell}}[q_\sigma^t(b) = x \mid \tau^{S-ab}(k) = x]$$

$$+ (1-p) \cdot \frac{1}{m-2} \cdot \sum_{k=\max(\ell-t,1)}^{\ell-1} Pr_{\sigma \sim D_{i,j,\ell}}[q_\sigma^t(b) = x \mid \tau^{S-ab}(k) = x]$$

$$= p \cdot \frac{1}{m-2} \cdot \sum_{k=\max(j-t,1)-1}^{j-2} p_{j,k+1}^t$$

$$+ (1-p) \cdot \frac{1}{m-2} \cdot \sum_{k=\max(\ell-t,1)}^{\ell-1} p_{\ell,k}^t$$

$$= p \cdot \frac{1}{m-2} + (1-p) \cdot \frac{1}{m-2} = \frac{1}{m-2}.$$

Next, we prove that for $p = \frac{P_\ell^t}{P_i^t - P_j^t + P_\ell^t}$, and every $x \in S_{-ab}$, it holds that $Pr_{\sigma \sim D_{i,j,\ell}}[q_\sigma^t(x) = a] = Pr_{\sigma \sim D_{i,j,\ell}}[q_\sigma^t(x) = b]$. Note that

$$Pr_{\sigma \sim D_{i,j,\ell}}[q_\sigma^t(x) = a] = p \cdot \sum_{k=i}^{i+t-1} Pr[\tau^{S-ab}(k) = x] \cdot Pr_{\sigma \sim D_{i,j,\ell}}[q_\sigma^t(x) = a \mid \tau^{S-ab}(k) = x]$$

$$= p \cdot \frac{1}{m-2} \cdot \sum_{k=i}^{i+t-1} Pr_{\sigma \sim D_{i,j,\ell}}[q_\sigma^t(x) = a \mid \tau^{S-ab}(k) = x]$$

$$= p \cdot \frac{1}{m-2} \cdot \sum_{k=i}^{i+t-1} p_{k+1,i}^t$$

$$= \frac{p}{m-2} \cdot P_i^t$$

where the last equality is by definition of $P_i^t$.

Similarly,

$$Pr_{\sigma \sim D_{i,j,\ell}}[q_\sigma^t(x) = b] = p \cdot \sum_{k=j-1}^{j+t-2} Pr[\tau^{S-ab}(k) = x] Pr_{\sigma \sim D_{i,j,\ell}}[q_\sigma^t(x) = b \mid \tau^{S-ab}(k) = x]$$

$$+ (1-p) \cdot \sum_{k=\ell}^{\ell+t-1} Pr[\tau^{S-ab}(k) = x] \cdot Pr_{\sigma \sim D_{i,j,\ell}}[q_\sigma^t(x) = b \mid \tau^{S-ab}(k) = x]$$

$$= \frac{p}{m-2} \cdot \sum_{k=j-1}^{j+t-2} Pr_{\sigma \sim D_{i,j,\ell}}[q_\sigma^t(x) = b \mid \tau^{S-ab}(k) = x]$$

$$+ \frac{1-p}{m-2} \cdot \sum_{k=\ell}^{\ell+t-1} Pr_{\sigma \sim D_{i,j,\ell}}[q_\sigma^t(x) = b \mid \tau^{S-ab}(k) = x]$$

$$= \frac{p}{m-2} \cdot \sum_{k=j-1}^{j+t-2} p_{k+2,j}^t + \frac{1-p}{m-2} \cdot \sum_{k=\ell}^{\ell+t-1} p_{k+1,\ell}^t$$

$$= \frac{p}{m-2} \cdot P_j^t + \frac{1-p}{m-2} \cdot P_\ell^t.$$

Observe that we can make the two probabilities above equal by setting $p = \frac{P_\ell^t}{P_i^t - P_j^t + P_\ell^t}$. Lastly, for any $x, y \in S_{-ab}$ it holds that $Pr_{\sigma \sim D_{i,j,\ell}}[q_\sigma^t(x) = y] = Pr_{\sigma \sim D_{i,j,\ell}^{a \leftrightarrow b}}[q_\sigma^t(x) = y]$, since the two preference profiles differ only with respect to $a, b$. From all the above, we get that $D_{i,j,\ell}$ and $D_{i,j,\ell}^{a \leftrightarrow b}$ are $t$-indistinguishable by setting $p$ as stated in the lemma.

$\square$

## B. Proof of Lemma 4.1

*Proof.* Given $D$, where, without loss of generality, $sc_{\vec{s}}(a, D) > sc_{\vec{s}}(b, D)$, we construct a family of preference profiles $\{D^c\}_{c \in M}$ in the same way as in Lemma 4.1 of Halpern et al. (2024). Briefly, we sample a permutation $\pi$ over the candidates uniformly at random. If $\pi(b) = c$, we sample a ranking from $\pi \circ D^{a \leftrightarrow b}$; otherwise, if $\pi(b) \neq c$, we sample a ranking from $\pi \circ D$. When $sc_{\vec{s}}(a, D) > sc_{\vec{s}}(b, D)$, Halpern et al. (2024) show in Lemma 4.1 that $c$ is the unique score maximizer in $D^c$.

It remains to show that $\{D^c\}_{c \in M}$ are $t$-indistinguishable from one another. Note that if $D$ and $D^{a \leftrightarrow b}$ are $t$-indistinguishable, then $\pi \circ D$ and $\pi \circ D^{a \leftrightarrow b}$ are also $t$-indistinguishable for all $\pi$. Consequently, for every $D^c$ and $D^{c'}$ and for all $x, y \in M$, we have:

$$\Pr_{\sigma \sim D^c}[q_\sigma^t(x) = y] = \sum_{\pi \in \mathcal{L}(M): \pi(b) \neq c} \Pr_{\sigma \sim \pi \circ D}[q_\sigma^t(x) = y] \cdot \frac{1}{m!} + \sum_{\pi \in \mathcal{L}(M): \pi(b) = c} \Pr_{\sigma \sim \pi \circ D^{a \leftrightarrow b}}[q_\sigma^t(x) = y] \cdot \frac{1}{m!}$$

$$= \frac{1}{m!} \sum_{\pi \in \mathcal{L}(M)} \Pr_{\sigma \sim \pi \circ D}[q_\sigma^t(x) = y]$$

$$= \sum_{\pi \in \mathcal{L}(M): \pi(b) \neq c'} \Pr_{\sigma \sim \pi \circ D}[q_\sigma^t(x) = y] \cdot \frac{1}{m!} + \sum_{\pi \in \mathcal{L}(M): \pi(b) = c'} \Pr_{\sigma \sim \pi \circ D^{a \leftrightarrow b}}[q_\sigma^t(x) = y] \cdot \frac{1}{m!}$$

$$= \Pr_{\sigma \sim D^{c'}}[q_\sigma^t(x) = y],$$

where the first and third equalities follow because $\pi$ is chosen uniformly at random from $\mathcal{L}(M)$ and the second and fourth equalities follow from the fact that $\pi \circ D$ and $\pi \circ D^{a \leftrightarrow b}$ are $t$-indistinguishable, and the lemma follows.

$\square$

## C. Proof of Lemma 4.3

*Proof.* First, we note that $sc_{s_{plu}}(a, D) = \Pr_{\sigma \sim D}[\sigma^{-1}(a) = 1] = \Pr_{\sigma \sim D}[q^t(a) = a]$, where the second equality is derived from the fact that when $a$ is the top choice of an agent the agent just returns this candidate. Since the algorithm knows $\Pr_{\sigma \sim D}[q^t(a) = a]$, the plurality score of candidate $a$ is computable.

Second, note that

$$\sum_{b \in M \backslash \{a\}} \Pr_{\sigma \sim D}[q^t_\sigma(b) = a] =$$

$$= \sum_{b \in M \backslash \{a\}} \sum_{i=1}^{m} \sum_{j=i+1}^{\min(m,i+t)} p^t_{j,i} \cdot \Pr_{\sigma \sim D}[\sigma^{-1}(a) = i, \sigma^{-1}(b) = j]$$

$$= \sum_{i=1}^{m} \sum_{j=i+1}^{\min(m,i+t)} \sum_{b \in M \backslash \{a\}} p^t_{j,i} \cdot \Pr_{\sigma \sim D}[\sigma^{-1}(a) = i, \sigma^{-1}(b) = j]$$

$$= \sum_{i=1}^{m} \sum_{j=i+1}^{\min(m,i+t)} \sum_{b \in M \backslash \{a\}} p^t_{j,i} \cdot \Pr_{\sigma \sim D}[\sigma^{-1}(a) = i \mid \sigma^{-1}(b) = j] \cdot \Pr_{\sigma \sim D}[\sigma^{-1}(b) = j]$$

$$= \sum_{i=1}^{m} \sum_{j=i+1}^{\min(m,i+t)} p^t_{j,i} \cdot \Pr_{\sigma \sim D}[\sigma^{-1}(a) = i]$$

$$= \sum_{i=1}^{m} P^t_i \cdot \Pr_{\sigma \sim D}[\sigma^{-1}(a) = i] = sc_{\vec{s}^*_t}(a, D).$$

where the first equality follows from the definition of $\Pr_{\sigma \sim D}[q^t_\sigma(b) = a]$, the second equality follows by rearranging the summations, the second last equality follows from the definition of $P^t_i$ and the last equality follows from the definition of $\vec{s}^*_t$. Since the algorithm knows the quantity $Pr_{\sigma \sim D}[q^t_\sigma(b) = a]$ for all $a, b \in M$, it can learn $sc_{\vec{s}^*_t}(a, D)$ for every $a \in M$.

The lemma follows by noticing that for each $\vec{s} = \lambda_1 \cdot \vec{s}_{plu} + \lambda_2 \cdot \vec{s}^*_t$, from linearity of expectation we have that $sc_{\vec{s}}(a, D) = \lambda_1 \cdot sc_{\vec{s}_{plu}}(a, D) + \lambda_2 \cdot sc_{\vec{s}^*_t}(a, D)$ for any $\lambda_1$ and $\lambda_2$. $\square$

*Proof.* Let $a, b$ be any two candidates and $\vec{s} = (s_1, \ldots, s_m)$ be any positional scoring rule.

We will first define an additional family of preference profiles, denoted $\hat{D}_i$, which will be used to prove impossibility results for the uniform $t$-improvement distribution.

**Lemma C.1.** *Let $\hat{D}_i$ be preference profiles that are defined with respect to two candidates $a$ and $b$, as follows:*

1. *With probability $p$, candidate $a$ is fixed at position $i$, and candidate $b$ is fixed at position $i + 1$, where $max(2, m - t) \leq i < m - 1$.*

2. *With probability $1 - p$, candidate $a$ is fixed at position $m$, and candidate $b$ is fixed at position $m - 1$.*

3. *Select a uniformly random ranking $\tau^{S-ab}$ over the set of all remaining candidates (excluding $a$ and $b$). This ranking determines the relative order of the remaining candidates, which are then placed in the positions not occupied by $a$ or $b$.*

*For any $t \in [m - 1]$, in the uniform $t$-improvement feedback distribution, the preference profiles $\hat{D}_i$ and $\hat{D}_i^{a \leftrightarrow b}$ are $t$-indistinguishable, when $p = \frac{min(i,t)}{t+min(i,t)}$.*

*Proof.* Let $a, b$ be the candidates of the statement.

Firstly, we need to show that $\Pr_{\sigma \sim \hat{D}_i}[q^t_\sigma(a) = b] = \Pr_{\sigma \sim \hat{D}_i}[q^t_\sigma(b) = a]$. Under the uniform $t$-improvement feedback distribution, with probability $p$, candidate $a$ appears immediately above candidate $b$ in the ranking. In this case, candidate $b$

returns candidate $a$ with uniform probability over the $t$-above neighborhood positions, which is $\frac{1}{\min(i,t)}$. Similarly, with probability $1 - p$, candidate $b$ appears immediately above candidate $a$ in the ranking. In this case, candidate $a$ returns candidate $b$ over the $t$-above neighborhood positions, which is $\frac{1}{t}$. By construction of $\hat{D}_i$, these two probabilities are equal for the choice of $p = \min(i,t)/(t + \min(i,t))$.

Now, notice that when querying a candidate $x \in S_{-ab}$, the agent can return $a$ or $b$ only if $x$ is ranked at some $j > i+1$ when $a$ is at position $i$ and $b$ at position $i+1$. Since $i \geq \max(2, m - t)$, under the uniform $t$-improvement feedback distribution, querying any candidate $x \in S_{-ab}$ ranked at position $j > i+1$ results in the agent returning $a$ or $b$ with equal probability, which is $1/\min(j-1, t)$. Hence, $\Pr_{\sigma \sim \hat{D}_i}[q^t_\sigma(x) = a] = \Pr_{\sigma \sim \hat{D}_i}[q^t_\sigma(x) = b]$.

Moreover,

$$
\begin{aligned}
Pr_{\sigma \sim \hat{D}_i}[q^t_\sigma(b) = x] &= p \cdot \sum_{k=i-min(i,t)+1}^{i-1} Pr[\tau^{S_{-ab}}(k) = x] \cdot Pr_{\sigma \sim \hat{D}_i}[q^t_\sigma(b) = x \mid \tau^{S_{-ab}}(k) = x] \\
&+ (1-p) \cdot \sum_{k=m-t-1}^{m-2} Pr[\tau^{S_{-ab}}(k) = x] \cdot Pr_{\sigma \sim \hat{D}_i}[q^t_\sigma(b) = x \mid \tau^{S_{-ab}}(k) = x] \\
&= p \cdot \frac{1}{m-2} \cdot \sum_{k=i-min(i,t)+1}^{i-1} Pr_{\sigma \sim \hat{D}_i}[q^t_\sigma(b) = x \mid \tau^{S_{-ab}}(k) = x] \\
&+ (1-p) \cdot \frac{1}{m-2} \cdot \sum_{k=m-t-1}^{m-2} Pr_{\sigma \sim \hat{D}_i}[q^t_\sigma(b) = x \mid \tau^{S_{-ab}}(k) = x] \\
&= p \cdot \frac{1}{m-2} \cdot \sum_{k=i-min(i,t)+1}^{i-1} p^t_{i+1,k} + (1-p) \cdot \frac{1}{m-2} \cdot \sum_{k=m-t-1}^{m-2} p^t_{m-1,k} \\
&= p \cdot \frac{1}{m-2} \cdot \frac{min(i,t)-1}{min(i,t)} + (1-p) \cdot \frac{1}{m-2}
\end{aligned}
$$

where the second equality holds because $\tau^{S_{-ab}}$ is chosen uniformly at random and the last equality holds since each agent returns every candidate in the $t$-above neighborhood with the same probability.

Similarly,

$$
\begin{aligned}
Pr_{\sigma \sim \hat{D}_i}[q^t_\sigma(a) = x] &= p \cdot \sum_{k=i-min(i-1,t)}^{i-1} Pr[\tau^{S_{-ab}}(k) = x] \cdot Pr_{\sigma \sim \hat{D}_i}[q^t_\sigma(a) = x \mid \tau^{S_{-ab}}(k) = x] \\
&+ (1-p) \cdot \sum_{k=m-t}^{m-2} Pr[\tau^{S_{-ab}}(k) = x] \cdot Pr_{\sigma \sim \hat{D}_i}[q^t_\sigma(a) = x \mid \tau^{S_{-ab}}(k) = x] \\
&= p \cdot \frac{1}{m-2} \cdot \sum_{k=i-min(i-1,t)}^{i-1} Pr_{\sigma \sim \hat{D}_i}[q^t_\sigma(a) = x \mid \tau^{S_{-ab}}(k) = x] \\
&+ (1-p) \cdot \frac{1}{m-2} \cdot \sum_{k=m-t}^{m-2} Pr_{\sigma \sim \hat{D}_i}[q^t_\sigma(a) = x \mid \tau^{S_{-ab}}(k) = x] \\
&= p \cdot \frac{1}{m-2} \cdot \sum_{k=i-min(i-1,t)}^{i-1} p^t_{i,k} + (1-p) \cdot \frac{1}{m-2} \cdot \sum_{k=m-t}^{m-2} p^t_{m,k} \\
&= p \cdot \frac{1}{m-2} + (1-p) \cdot \frac{1}{m-2} \cdot \frac{t-1}{t}.
\end{aligned}
$$

Therefore, we have $Pr_{\sigma \sim \hat{D}_i}[q_\sigma^t(a) = x] = Pr_{\sigma \sim \hat{D}_i}[q_\sigma^t(b) = x]$, for the choice of $p = \min(i, t)/(t + \min(i, t))$.

Lastly, for any $x, y \in S_{-ab}$ it holds that $Pr_{\sigma \sim \hat{D}_i}[q_\sigma^t(x) = y] = Pr_{\sigma \sim \hat{D}_i^{a \leftrightarrow b}}[q_\sigma^t(x) = y]$, since the two preference profiles differ only with respect to $a, b$. From all the above, we get that $\hat{D}_i$ and $\hat{D}_i^{a \leftrightarrow b}$ are $t$-indistinguishable by setting $p$ as stated in the lemma. $\qquad \square$

We will continue with the following three lemmas.

## D. Proof of Lemma 4.5

*Proof.* Consider the preference profile $D_{i,m,i+1}$ as stated in Definition 3.1. From Lemma 3.2, we get that for every $i \in \{2, \ldots, m - t - 2\}$, $D_{i,m,i+1}$ and $D_{i,m,i+1}^{a \leftrightarrow b}$ are $t$-indistinguishable for every value of $t \leq m - 4$, if $p = \frac{P_{i+1}^t}{P_i^t - P_m^t + P_{i+1}^t} = \frac{P_{i+1}^t}{P_i^t + P_{i+1}^t}$. If it holds that $sc_{\vec{s}}(a, D_{i,m,i+1}) > sc_{\vec{s}}(b, D_{i,m,i+1})$, for some $i \in \{2, \ldots, m - t - 2\}$, then by setting $D = D_{i,m,i+1}$, the statement of the lemma is satisfied. Now, suppose that for every $i \in \{2, \ldots, m - t - 2\}$, it holds that $sc_{\vec{s}}(a, D_{i,m,i+1}) = sc_s(b, D_{i,m,i+1})$. This means that

$$p \cdot s_i = (1 - p) \cdot s_{i+1} \implies p = \frac{s_{i+1}}{s_i + s_{i+1}}.$$

Therefore, we get

$$p = \frac{P_{i+1}^t}{P_i^t + P_{i+1}^t} = \frac{s_{i+1}}{s_i + s_{i+1}} \implies \frac{s_{i+1}}{P_{i+1}^t} = \frac{s_i}{P_i^t} = \lambda$$

and as a result: $s_i/P_i^t = \lambda$ for all $i \in \{2, \ldots, m - t - 1\}$ and the lemma follows. $\qquad \square$

**Lemma D.1.** *If $s_i/P_i^t = \mu$ does not hold for all $i \in \{\max(2, m - t), \ldots, m - 1\}$ for some $\mu$, then there exists a preference profile $D$ such that $sc_{\vec{s}}(a, D) > sc_{\vec{s}}(b, D)$, and $D$ and $D^{a \leftrightarrow b}$ are $t$-indistinguishable, for every $t \in [m - 1]$. .*

*Proof.* Consider the preference profile $\hat{D}_i$ as stated in Lemma C.1. From Lemma C.1, we get that for *every* $i \in \{max(2, m - t), \ldots, m - 2\}$, $\hat{D}_i$ and $\hat{D}_i^{a \leftrightarrow b}$ are $t$-indistinguishable for every value of $t$, if $p = \frac{min(i,t)}{t + min(i,t)}$. Under uniform $t$-improvement feedback distribution, we know that $P_{m-1}^t = \frac{1}{t}$ and $P_i^t - P_{i+1}^t = \frac{1}{min(i,t)}$, for all $i \in \{max(2, m - t), \cdots, m - 2\}$. Hence, we have

$$\frac{P_{m-1}^t}{P_i^t - P_{i+1}^t + P_{m-1}^t} = \frac{1/t}{1/t + 1/min(i,t)} = \frac{min(i,t)}{t + min(i,t)} = p.$$

If for some $i \in \{max(2, m - t), \ldots, m - 2\}$, $sc_{\vec{s}}, (a, \hat{D}_i) > sc_{\vec{s}}(b, \hat{D}_i)$, then by setting $D = \hat{D}_i$, the statement of the lemma is satisfied. Now, suppose that for every $i \in \{max(2, m - t), \ldots, m - 2\}$, $sc_{\vec{s}}(a, D_{i,m,i+1}) = sc_{\vec{s}}(b, D_{i,m,i+1})$. This means that

$$p \cdot s_i = p \cdot s_{i+1} + (1 - p) \cdot s_{m-1} \implies p = \frac{s_{m-1}}{s_i - s_{i+1} + s_{m-1}}.$$

Therefore, we get

$$p = \frac{P_{m-1}^t}{P_i^t - P_{i+1}^t + P_{m-1}^t} = \frac{s_{m-1}}{s_i - s_{i+1} + s_{m-1}} \implies \frac{s_i - s_{i+1}}{s_{m-1}} = \frac{P_i^t - P_{i+1}^t}{P_{m-1}^t}.$$

Now, observe that, substituting $i = m - 2$ in the above, we get $\frac{s_{m-2} - s_{m-1}}{s_{m-1}} = \frac{P_{m-2}^t - P_{m-1}^t}{P_{m-1}^t}$ which implies that $\frac{s_{m-2}}{P_{m-2}^t} = \frac{s_{m-1}}{P_{m-1}^t}$. By induction on $i$, we get for all $i \in \{max(2, m - t), \ldots, m - 1\}$, that $\mu = \frac{s_i}{P_i^t}$ and the lemma follows. $\qquad \square$

**Lemma D.2.** *If $s_i/P_i^t = \lambda$ for all $i \in \{2, \ldots, m - t - 1\}$ for some $\lambda$ and $s_i/P_i^t = \mu$ for all $i \in \{\max(2, m - t), \ldots, m - 1\}$ for some $\mu$, and $\mu \neq \lambda$, then there exists a preference profile $D$ such that $sc_{\vec{s}}(a, D) > sc_{\vec{s}}(b, D)$, and $D$ and $D^{a \leftrightarrow b}$ are $t$-indistinguishable.*

*Proof.* First, we note that when $t = m - 1$ or $t = m - 2$, from the hypothesis of the lemma we directly get that $s_2^t/P_2^t = \ldots = s_{m-1}^t/P_{m-1}^t = \mu$. Therefore for these two cases the lemma directly follows. Next, we focus on the case where $t \leq m - 3$.

Here we consider the following preference profile, denoted by $D$:

1. With probability $p$, candidate $a$ is fixed at position 2, and candidate $b$ is fixed at position $m$.

2. With probability $(1-p)/2$, candidate $a$ is fixed at position 1, and candidate $b$ is fixed at position $m - 1$.

3. With probability $(1-p)/2$, candidate $a$ is fixed at position $m$, and candidate $b$ is fixed at position 1.

4. Select a uniformly random ranking $\tau^{S-ab}$ over the set of all remaining candidates (excluding $a$ and $b$). This ranking determines the relative order of the remaining candidates, which are then placed in the positions not occupied by $a$ or $b$.

First, we will show that $D$ and $D^{a \leftrightarrow b}$ are $t$-indistinguishable. Since in all the cases $a, b$ are more than $m - 3$ positions apart, it holds that $Pr_{\sigma \sim D}[q_\sigma^t(b) = a] = Pr_{\sigma \sim D}[q_\sigma^t(a) = b] = 0$. Furthermore, $Pr_{\sigma \sim D}[q_\sigma^t(b) = b] = Pr_{\sigma \sim D}[q_\sigma^t(a) = a] = \frac{1-p}{2}$. Next we note that for any $x \in S_{-ab}$, it holds that

$$Pr_{\sigma \sim D}[q_\sigma^t(a) = x] = p \cdot Pr[\tau^{S-ab}(1) = x] \cdot Pr_{\sigma \sim D}[q_\sigma^t(a) = x \mid \tau^{S-ab}(1) = x]$$

$$+ \frac{(1-p)}{2} \cdot \sum_{k=m-t-1}^{m-2} Pr[\tau^{S-ab}(k) = x] \cdot Pr_{\sigma \sim D}[q_\sigma^t(a) = x \mid \tau^{S-ab}(k) = x]$$

$$= p \cdot \frac{1}{m-2} \cdot Pr_{\sigma \sim D}[q_\sigma^t(a) = x \mid \tau^{S-ab}(1) = x]$$

$$+ \frac{(1-p)}{2} \cdot \frac{1}{m-2} \sum_{k=m-t-1}^{m-2} Pr_{\sigma \sim D}[q_\sigma^t(a) = x \mid \tau^{S-ab}(k) = x]$$

$$= p \cdot p_{2,1}^t \cdot \frac{1}{m-2} + \frac{(1-p)}{2} \cdot \frac{1}{m-2} \cdot \sum_{k=m-t-1}^{m-2} p_{m,k+1}^t$$

$$= p \cdot \frac{1}{m-2} + \frac{(1-p)}{2(m-2)} = \frac{p+1}{2(m-2)}$$

where the second equality comes from the fact that $\tau^{S-ab}$ is picked uniformly at random and the third equality by the fact that each agent returns a candidate from the $t$-above neighborhood.

Similarly, for any $x \in S_{-ab}$, it holds that

$$Pr_{\sigma \sim D}[q_\sigma^t(b) = x] = p \cdot \sum_{k=m-t-1}^{m-2} Pr[\tau^{S-ab}(k) = x] \cdot Pr_{\sigma \sim D}[q_\sigma^t(b) = x \mid \tau^{S-ab}(k) = x]$$

$$+ \frac{(1-p)}{2} \cdot \sum_{k=m-t-2}^{m-3} Pr[\tau^{S-ab}(k) = x] \cdot Pr_{\sigma \sim D}Pr[q_\sigma^t(b) = x \mid \tau^{S-ab}(k) = x]$$

$$= p \cdot \frac{1}{m-2} \cdot \sum_{k=m-t-1}^{m-2} Pr_{\sigma \sim D}[q_\sigma^t(b) = x \mid \tau^{S-ab}(k) = x]$$

$$+ \frac{(1-p)}{2} \cdot \frac{1}{m-2} \sum_{k=m-t-2}^{m-3} Pr_{\sigma \sim D}[q_\sigma^t(b) = x \mid \tau^{S-ab}(k) = x]$$

$$= p \cdot \frac{1}{m-2} \cdot \sum_{k=m-t-1}^{m-2} p_{m,k+1}^t + \frac{(1-p)}{2} \cdot \frac{1}{m-2} \sum_{k=m-t-2}^{m-3} p_{m-1,k+1}^t$$

$$= p \cdot \frac{1}{m-2} + \frac{(1-p)}{2(m-2)} = \frac{p+1}{2(m-2)}.$$

Therefore, we have that $Pr_{\sigma\sim D}[q_\sigma^t(a) = x] = Pr_{\sigma\sim D}[q_\sigma^t(b) = x]$ for all $x \in S_{-ab}$. Next, we see that

$$Pr_{\sigma\sim D}[q_\sigma^t(x) = a] = p \cdot \sum_{k=2}^{t+1} Pr[\tau^{S_{-ab}}(k) = x] \cdot Pr_{\sigma\sim D}[q_\sigma^t(x) = a \mid \tau^{S_{-ab}}(k) = x]$$

$$+ \frac{1-p}{2} \cdot \sum_{k=1}^{t} Pr[\tau^{S_{-ab}}(k) = x] \cdot Pr_{\sigma\sim D}[q_\sigma^t(x) = a \mid \tau^{S_{-ab}}(k) = x]$$

$$= p \cdot \frac{1}{m-2} \cdot \sum_{k=2}^{t+1} \cdot Pr_{\sigma\sim D}[q_\sigma^t(x) = a \mid \tau^{S_{-ab}}(k) = x]$$

$$+ \frac{1-p}{2} \cdot \frac{1}{m-2} \cdot \sum_{k=1}^{t} \cdot Pr_{\sigma\sim D}[q_\sigma^t(x) = a \mid \tau^{S_{-ab}}(k) = x]$$

$$= p \cdot \frac{1}{m-2} \sum_{k=2}^{t+1} p_{k+1,2}^t + \frac{1-p}{2} \cdot \frac{1}{m-2} \cdot \sum_{k=1}^{t} p_{k+1,1}^t$$

$$= p \cdot \frac{1}{m-2} \cdot P_2^t + \frac{1-p}{2} \cdot \frac{1}{m-2} P_1^t$$

where the second equality comes from the fact that $\tau^{S_{-ab}}$ is picked uniformly at random and the last equality by definition of $P_i^t$. Similarly,

$$Pr_{\sigma\sim D}[q_\sigma^t(x) = b] = \frac{1-p}{2} \cdot \sum_{k=1}^{t} Pr[\tau^{S_{-ab}}(k) = x] \cdot Pr_{\sigma\sim D}[q_\sigma^t(x) = b \mid \tau^{S_{-ab}}(k) = x]$$

$$+ \frac{1-p}{2} \cdot Pr[\tau^{S_{-ab}}(m-2) = x] \cdot Pr_{\sigma\sim D}[q_\sigma^t(x) = b \mid \tau^{S_{-ab}}(m-2) = x]$$

$$= \frac{1-p}{2} \cdot \frac{1}{m-2} \cdot \sum_{k=1}^{t} Pr_{\sigma\sim D}[q_\sigma^t(x) = b \mid \tau^{S_{-ab}}(k) = x]$$

$$+ \frac{1-p}{2} \cdot \frac{1}{m-2} Pr_{\sigma\sim D}[q_\sigma^t(x) = b \mid \tau^{S_{-ab}}(m-2) = x]$$

$$= \frac{1-p}{2} \cdot \frac{1}{m-2} \sum_{k=1}^{t} p_{k+1,1}^t + \frac{1-p}{2} \cdot \frac{1}{m-2} \cdot p_{m,m-1}^t$$

$$= \frac{1-p}{2} \cdot \frac{1}{m-2} \cdot P_1^t + \frac{1-p}{2} \cdot \frac{1}{m-2} P_{m-1}^t$$

Therefore, we get $Pr_{\sigma\sim D}[q_\sigma^t(x) = a] = Pr_{\sigma\sim D}[q_\sigma^t(x) = b]$ when $p = P_{m-1}^t/(2P_2^t + P_{m-1}^t)$ and then $D$ and $D^{a\leftrightarrow b}$ become $t$-indistinguishable.

If it holds that $sc_{\vec{s}}(a, D) \neq sc_{\vec{s}}(b, D)$, then the statement of the lemma is satisfied. Now, suppose that $sc_{\vec{s}}(a, D) = sc_{\vec{s}}(b, D)$. This means that

$$p \cdot s_2 + \frac{1-p}{2} \cdot s_1 = \frac{1-p}{2} \cdot s_1 + \frac{1-p}{2} \cdot s_{m-1} \implies p = \frac{s_{m-1}}{2 \cdot s_2 + s_{m-1}} = \frac{P_{m-1}^t}{2 \cdot P_2^t + P_{m-1}^t}.$$

For $t \leq m - 4$ we know that $s_2/P_2^t = \lambda$ and $s_{m-1}/P_{m-1}^t = \mu$. Then from the above equality we get $\mu = \lambda$. For $t = m - 3$ we know that $s_3/P_3^t = \ldots = s_{m-1}/P_{m-1}^t = \mu$ and the above implies that $s_2/P_2^t = s_3/P_3^t = \ldots = s_{m-1}/P_{m-1}^t = \mu$, and the lemma follows.

$\square$

From the above lemma we get that, unless $\frac{s_i}{P_i^t} = \lambda$ for all $i \in \{2, \ldots, m-1\}$, which implies that $\vec{s} \in \text{span}(\vec{s}_{\text{plu}}, \vec{s}_t^*)$, there

exists a preference profile $D$ satisfying the conditions of the lemma and the lemma follows.

# E. Proof of Lemma 5.1

*Proof.* We use the same construction as in Lemma 4.1 by Halpern et al. (2024), which we briefly restate here for completeness. For each $c \in M$ we define $D^c$ as following: we pick a permutation $\pi$ uniformly at random. If $\pi(b) = c$ then we sample a ranking $\sigma \sim \pi \circ D^{a \leftrightarrow b}$. If $\pi(b) \neq c$ then we sample a ranking $\sigma \sim \pi \circ D$. We need to show that the two following statements are true: (1) $Pr_{\sigma \sim D^c}[c \succ_\sigma x] > Pr_{\sigma \sim D^c}[x \succ_\sigma c]$, for all $x \in M \setminus \{c\}$ and (2) $\{D^c\}_{c \in M}$ consists of $m$ t-indistinguishable preference profiles. Firstly, we turn our attention to (1).

Fix any $x \in M \setminus \{c\}$. First note that,

$$\sum_{z \in \{c,x\}} \Pr_{\sigma \sim D^c}[c \succ_\sigma x \mid \pi(b) = z] \cdot Pr[\pi(b) = z]$$

$$= \frac{1}{m} \cdot \left( Pr_{\sigma \sim D}[a \succ_\sigma \pi^{ab}(\pi^{-1}(x)) \mid \pi(b) = z] + Pr_{\sigma \sim D}[\pi^{-1}(c) \succ_\sigma b \mid \pi(b) = z] \right)$$

$$= \frac{1}{m} \left( \sum_{y \in S_{-ab}} Pr_{\sigma \sim D}[a \succ_\sigma y \mid \pi^{-1}(x) = y, \pi(b) = z] \cdot Pr[\pi^{-1}(x) = y \mid \pi(b) = z] \right.$$

$$+ Pr_{\sigma \sim D}[a \succ_\sigma b \mid \pi^{-1}(x) = a, \pi(b) = z] \cdot Pr[\pi^{-1}(x) = a \mid \pi(b) = z]$$

$$+ \sum_{y \in S_{-ab}} Pr_{\sigma \sim D}[y \succ_\sigma b \mid \pi^{-1}(c) = y, \pi(b) = z] \cdot Pr[\pi^{-1}(c) = y \mid \pi(b) = z]$$

$$\left. + Pr_{\sigma \sim D}[a \succ_\sigma b \mid \pi^{-1}(c) = a, \pi(b) = z] \cdot Pr[\pi^{-1}(c) = a \mid \pi(b) = z] \right)$$

$$= \frac{1}{m \cdot (m-1)} \left( \sum_{x \in M \setminus \{a\}} Pr_{\sigma \sim D}[a \succ_\sigma x] + \sum_{x \in M \setminus \{b\}} Pr_{\sigma \sim D}[x \succ_\sigma b] \right)$$

$$= \frac{1}{m \cdot (m-1)} \left( \sum_{x \in M \setminus \{a\}} Pr_{\sigma \sim D}[a \succ_\sigma x] + \sum_{x \in M \setminus \{b\}} (1 - Pr_{\sigma \sim D}[b \succ_\sigma x]) \right), \tag{2}$$

where the second equality holds, since $\pi$ is a permutation selected uniformly at random and conditioned on $\pi(b) = c$, $D^c = \pi \circ D^{a \leftrightarrow b}$ and conditioned on $\pi(b) = x$, $D^c = \pi \circ D$. The second last equality holds again from the fact that $\pi$ is a permutation selected uniformly at random and that $D$ is independent of $\pi$.

By the same reasoning we can also argue that,

$$\sum_{z \in \{c,x\}} \Pr_{\sigma \sim D^c}[x \succ_\sigma c \mid \pi(b) = z] \cdot Pr[\pi(b) = z]$$

$$= \frac{1}{m \cdot (m-1)} \left( \sum_{x \in M \setminus \{a\}} Pr_{\sigma \sim D}[x \succ_\sigma a] + \sum_{x \in M \setminus \{b\}} Pr_{\sigma \sim D}[b \succ_\sigma x] \right)$$

$$= \frac{1}{m \cdot (m-1)} \left( \sum_{x \in M \setminus \{a\}} (1 - Pr_{\sigma \sim D}[a \succ_\sigma x]) + \sum_{x \in M \setminus \{b\}} Pr_{\sigma \sim D}[b \succ_\sigma x] \right). \tag{3}$$

By combining Equation (2) and Equation (3), and the fact that $\sum_{x \in M \setminus \{a\}} Pr_{\sigma \sim D}[a \succ_\sigma x] > \sum_{x \in M \setminus \{b\}} Pr_{\sigma \sim D}[b \succ_\sigma x]$, we conclude that

$$\sum_{z \in \{c,x\}} \Pr_{\sigma \sim D^c}[c \succ_\sigma x \mid \pi(b) = z] \cdot Pr[\pi(b) = z] > \sum_{z \in \{c,x\}} \Pr_{\sigma \sim D^c}[x \succ_\sigma c \mid \pi(b) = z] \cdot Pr[\pi(b) = z]. \tag{4}$$

Next, note that

$$\sum_{z \in M \setminus \{c,x\}} Pr_{\sigma \sim D^c}[c \succ_\sigma x \mid \pi(b) = z] \cdot Pr[\pi(b) = z]$$

$$= \sum_{z \in M \setminus \{c,x\}} Pr_{\sigma \sim D}[\pi^{-1}(c) \succ_\sigma \pi^{-1}(x) \mid \pi(b) = z] \cdot \frac{1}{m}$$

$$= \frac{1}{m} \cdot \left( \sum_{y,y' \in M \setminus \{a\}} Pr_{\sigma \sim D}[y \succ_\sigma y' \mid \pi^{-1}(c) = y, \pi^{-1}(x) = y', \pi(b) = z] \right.$$

$$\cdot Pr[\pi^{-1}(c) = y, \pi^{-1}(x) = y' \mid \pi(b) = z]$$

$$+ \sum_{y \in M \setminus \{a\}} Pr_{\sigma \sim D}[y \succ_\sigma a \mid \pi^{-1}(c) = y, \pi^{-1}(x) = a, \pi(b) = z] \cdot Pr[\pi^{-1}(c) = y, \pi^{-1}(x) = a \mid \pi(b) = z]$$

$$\left. + \sum_{y \in M \setminus \{a\}} Pr_{\sigma \sim D}[a \succ_\sigma y \mid \pi^{-1}(c) = a, \pi^{-1}(x) = y, \pi(b) = z] \cdot Pr[\pi^{-1}(c) = a, \pi^{-1}(x) = y \mid \pi(b) = z] \right)$$

$$= \frac{1}{m(m-1)(m-2)} \cdot \left( \sum_{y,y' \in M \setminus \{a\}} Pr_{\sigma \sim D}[y \succ_\sigma y'] + \sum_{y \in M \setminus \{a\}} Pr_{\sigma \sim D}[y \succ_\sigma a] + \sum_{y \in M \setminus \{a\}} Pr_{\sigma \sim D}[a \succ_\sigma y] \right) \tag{5}$$

where the second equality holds, since $\pi$ is a permutation selected uniformly at random and conditioned on $\pi(b) \notin \{c, x\}$, $D^c = \pi \circ D$. The last equality holds again from the fact $\pi$ is a permutation selected uniformly at random and that $D$ is independent of $\pi$.

By the exact same reasoning, we get that

$$\sum_{z \in M \setminus \{c,x\}} Pr_{\sigma \sim D^c}[x \succ_\sigma c \mid \pi(b) = z] \cdot Pr[\pi(b) = z]$$

$$= \frac{1}{m(m-1)(m-2)} \cdot \left( \sum_{y,y' \in M \setminus \{a\}} Pr_{\sigma \sim D}[y \succ_\sigma y'] + \sum_{y \in M \setminus \{a\}} Pr_{\sigma \sim D}[a \succ_\sigma y] + \sum_{y \in M \setminus \{a\}} Pr_{\sigma \sim D}[y \succ_\sigma a] \right). \tag{6}$$

From Equation (5) and Equation (6), we get

$$\sum_{z \in M \setminus \{c,x\}} Pr_{\sigma \sim D^c}[c \succ_\sigma x \mid \pi(b) = z] \cdot Pr[\pi(b) = z] = \sum_{z \in M \setminus \{c,x\}} Pr_{\sigma \sim D^c}[x \succ_\sigma c \mid \pi(b) = z] \cdot Pr[\pi(b) = z]. \tag{7}$$

Lastly, by combining Equation (4) and Equation (7), we get $Pr_{\sigma \sim D^c}[c \succ_\sigma x] > Pr_{\sigma \sim D^c}[x \succ_\sigma c]$ as desired.

As for (2), we already have shown in Lemma 4.1 that the family of $\{D^c\}_{c \in M}$ consists of preference profiles that are all $t$-indistinguishable from one another. Hence, the lemma follows.

□

## F. Proof of Lemma 5.2

*Proof.* We will distinguish into the following cases:

**Case I: $2 < t < m - 3$.** Let $D_{3,m,2}$ be the preference profile defined in Lemma 3.2. Since $2 < t < m - 3$, for $p = \frac{P_2^t}{P_2^t + P_3^t}$, the preference profiles $D_{3,m,2}$ and $D_{3,m,2}^{a \leftrightarrow b}$ are $t$-indistinguishable.

Under $D_{3,m,2}$, we have $Pr_{\sigma \sim D_{3,m,2}}[a \succ_\sigma b] = p$ and $Pr_{\sigma \sim D_{3,m,2}}[b \succ_\sigma a] = 1 - p$. Additionally, for each $x \in S_{-ab}$,

we have that $\Pr_{\sigma \sim D_{3,m,2}}[a \succ_\sigma x] = p \cdot \frac{m-4}{m-2}$, since with probability $p$, $a$ appears at position 3, followed by $b$ at position $m$, and the remaining $m-2$ candidates are uniformly distributed across the remaining positions and with probability $(1-p)$, $a$ appears at the last position. Similarly, we have $\Pr_{\sigma \sim D_{3,m,2}}[b \succ_\sigma x] = (1-p) \cdot \frac{m-3}{m-2}$, since with probability $(1-p)$, $b$ appears at position 2, followed by $a$ at position $m$, and the remaining candidates are distributed uniformly, and with probability $p$, $b$ appears last.

Thus, we have that

$$\sum_{x \in M \setminus \{a\}} \Pr_{\sigma \sim D}[a \succ_\sigma x] = \sum_{x \in S_{-ab}} \Pr_{\sigma \sim D}[a \succ_\sigma x] + \Pr_{\sigma \sim D}[a \succ_\sigma b] = \sum_{x \in S_{-ab}} \frac{m-4}{m-2} \cdot p + p = (m-3) \cdot p,$$

and similarly, for $b$, we have:

$$\sum_{x \in M \setminus \{a\}} \Pr_{\sigma \sim D}[b \succ_\sigma x] = \sum_{x \in S_{-ab}} \Pr_{\sigma \sim D}[b \succ_\sigma x] + \Pr_{\sigma \sim D}[b \succ_\sigma a] = \sum_{x \in S_{-ab}} \frac{m-3}{m-2} \cdot (1-p) + (1-p) = (m-2) \cdot (1-p).$$

Next, we show that

$$p = \frac{P_2^t}{P_2^t + P_3^t} = \frac{P_2^t}{2P_2^t - \frac{1}{2} + \frac{1}{t}} > \frac{m-2}{2m-5},$$

where the inequality follows from the fact that, under the uniform $t$-improvement feedback distribution, for $t > 1$, we have that $P_3^t = P_2^t - \frac{1}{2} + \frac{1}{t}$. Then, this implies that $\sum_{x \in M \setminus \{a\}} \Pr_{\sigma \sim D}[a \succ_\sigma x] > \sum_{x \in M \setminus \{a\}} \Pr_{\sigma \sim D}[b \succ_\sigma x]$, as desired.

To ensure that $p > \frac{m-2}{2m-5}$, we need:

$$\frac{P_2^t}{2P_2^t - \frac{1}{2} + \frac{1}{t}} > \frac{m-2}{2m-5}$$
$$\implies 2m \cdot P_2^t - 5P_2^t > (m-2)(2P_2^t - 1/2 + 1/t)$$
$$\implies 2m \cdot P_2^t - 5P_2^t > 2m \cdot P_2^t - m/2 + m/t - 4P_2^t + 1 - 2/t$$
$$\implies P_2^t < \frac{m}{2} - \frac{m}{t} - 1 + \frac{2}{t} = (m-2)(\frac{1}{2} - \frac{1}{t})$$

Under the uniform $t$-improvement feedback distribution, we have:

$$P_2^t = \frac{1}{2} + \frac{1}{3} + \cdots + \frac{1}{t-1} + \frac{2}{t} = H_{t-1} - 1 + \frac{2}{t},$$

where $H_k$ is the $k$-th harmonic number. Using the inequality $H_k \leq \ln(k) + 1$, we get:

$$P_2^t \leq \ln(t-1) + \frac{2}{t}.$$

Since, $P_2^t \leq \ln(t-1) + \frac{2}{t}$, it suffices to prove that: $\ln(t-1) + \frac{2}{t} < (m-2)(\frac{1}{2} - \frac{1}{t})$ or $\ln(t-1) + \frac{m}{t} < \frac{m-2}{2}$.

We prove that for any value of $m \geq 13$ and $3 \leq t < m-3$, this is true, as follows:

We have that $\ln(t-1) + \frac{m}{t} < \ln(t) + \frac{m}{t}$. Let $f(t) = \ln(t) + \frac{m}{t}$. Then $f'(t) = 1/t - m/t^2$. Hence, $f'(m) = 0$ and for every $0 < t < m$, $f'(t) < 0$. Therefore, $f$ is monotone decreasing in $(0, m)$. As a result, $f(t) < f(3) = \ln(3) + m/3$. Now we need to find $m$ such that $\ln(3) + \frac{m}{3} < \frac{m-2}{2}$, or $m > 6 \cdot \ln(3) + 6$, or $m \geq 13$. This concludes the proof of the case.

**Case II:** $t \in \{1, 2\}$. We define the preference profile $D$ as following:

1. With probability $p$, candidate $a$ is fixed at position 2, and candidate $b$ is fixed at position 5.

2. With probability $1 - p$, candidate $a$ is fixed at position 6, and candidate $b$ is fixed at position 3.

3. Select a uniformly random ranking $\tau^{S-ab}$ over the set of all remaining candidates (excluding $a$ and $b$). This ranking determines the relative order of the remaining candidates, which are then placed in the positions not occupied by $a$ or $b$.

First, we show that $D$ and $D^{a \leftrightarrow b}$ are $t$-indistinguishable. Note that since $t \le 2$, we get that $Pr_{\sigma \sim D}[q_\sigma^t(a) = b] = Pr_{\sigma \sim D}[q_\sigma^t(b) = a] = 0$, since $a, b$ are more than $t$ positions apart. Furthermore, for each $x \in S_{-ab}$, we have:

$$
\begin{aligned}
&Pr_{\sigma \sim D}[q_\sigma^t(a) = x] \\
&= p \cdot Pr[\tau^{S-ab}(1) = x] \cdot Pr_{\sigma \sim D}[q_\sigma^t(a) = x \mid \tau^{S-ab}(1) = x] \\
&\quad + (1-p) \cdot \sum_{k=5-t}^{4} Pr[\tau^{S-ab}(k) = x] \cdot Pr_{\sigma \sim D}[q_\sigma^t(a) = x \mid \tau^{S-ab}(k) = x] \\
&= p \cdot \frac{1}{m-2} \cdot Pr_{\sigma \sim D}[q_\sigma^t(a) = x \mid \tau^{S-ab}(1) = x] + \frac{1-p}{m-2} \cdot \sum_{k=5-t}^{4} Pr_{\sigma \sim D}[q_\sigma^t(a) = x \mid \tau^{S-ab}(k) = x] \\
&= p \cdot \frac{1}{m-2} \cdot p_{2,1}^t + \frac{1-p}{m-2} \cdot \sum_{k=5-t}^{4} p_{6,k+1}^t \\
&= \frac{p}{m-2} + \frac{1-p}{m-2} = \frac{1}{m-2},
\end{aligned}
$$

where the first equality is because $\tau^{S-ab}$ is sampled uniformly at random and the last equality is since each agent returns a candidate in the $t$-above neighborhood of the suggested candidate.

Similarly,

$$
\begin{aligned}
&Pr_{\sigma \sim D}[q_\sigma^t(b) = x] \\
&= p \cdot \sum_{k=4-t}^{3} Pr[\tau^{S-ab}(k) = x] \cdot Pr_{\sigma \sim D}[q_\sigma^t(b) = x \mid \tau^{S-ab}(k) = x] \\
&\quad + (1-p) \cdot \sum_{k=3-t}^{2} Pr[\tau^{S-ab}(k) = x] \cdot Pr_{\sigma \sim D}[q_\sigma^t(b) = x \mid \tau^{S-ab}(k) = x] \\
&= \frac{p}{m-2} \cdot \sum_{k=4-t}^{3} Pr_{\sigma \sim D}[q_\sigma^t(b) = x \mid \tau^{S-ab}(k) = x] + \frac{1-p}{m-2} \cdot \sum_{k=3-t}^{2} Pr_{\sigma \sim D}[q_\sigma^t(b) = x \mid \tau^{S-ab}(k) = x] \\
&= \frac{p}{m-2} \cdot \sum_{k=4-t}^{3} p_{5,k+1}^t + \frac{1-p}{m-2} \cdot \sum_{k=3-t}^{2} p_{3,k}^t \\
&= \frac{p}{m-2} + \frac{1-p}{m-2} = \frac{1}{m-2}.
\end{aligned}
$$

Furthermore,

$$Pr_{\sigma \sim D}[q_\sigma^t(x) = a]$$

$$= p \cdot \sum_{k=2}^{1+t} Pr[\tau^{S-ab}(k) = x] \cdot Pr_{\sigma \sim D}[q_\sigma^t(x) = a \mid \tau^{S-ab}(k) = x]$$

$$+ (1-p) \cdot \sum_{k=5}^{min(m,4+t)} Pr[\tau^{S-ab}(k) = x] \cdot Pr_{\sigma \sim D}[q_\sigma^t(x) = a \mid \tau^{S-ab}(k) = x]$$

$$= \frac{p}{m-2} \sum_{k=2}^{1+t} Pr_{\sigma \sim D}[q_\sigma^t(x) = a \mid \tau^{S-ab}(k) = x] + \frac{1-p}{m-2} \sum_{k=5}^{min(m,4+t)} Pr_{\sigma \sim D}[q_\sigma^t(x) = a \mid \tau^{S-ab}(k) = x]$$

$$= \frac{1}{m-2} \cdot \left( p \cdot P_2^t + (1-p) \cdot P_6^t \right),$$

where the last equality is by definition of $P_2^t$ and the second to last is from the fact that $\tau^{S-ab}$ is picked uniformly at random.
Similarly,

$$Pr_{\sigma \sim D}[q_\sigma^t(x) = b] = p \cdot \sum_{k=4}^{min(3+t,m)} Pr[\tau^{S-ab}(k) = x] \cdot Pr_{\sigma \sim D}[q_\sigma^t(x) = b \mid \tau^{S-ab}(k) = x]$$

$$+ (1-p) \cdot \sum_{k=3}^{2+t} Pr[\tau^{S-ab}(k) = x] \cdot Pr_{\sigma \sim D}[q_\sigma^t(x) = b \mid \tau^{S-ab}(k) = x]$$

$$= \frac{p}{m-2} \sum_{k=4}^{min(3+t,m)} Pr_{\sigma \sim D}[q_\sigma^t(x) = b \mid \tau^{S-ab}(k) = x]$$

$$+ \frac{1-p}{m-2} \sum_{k=3}^{2+t} Pr_{\sigma \sim D}[q_\sigma^t(x) = b \mid \tau^{S-ab}(k) = x]$$

$$= \frac{1}{m-2} \cdot \left( p \cdot P_5^t + (1-p) \cdot P_3^t \right).$$

For getting $Pr_{\sigma \sim D}[q_\sigma^t(x) = a] = Pr_{\sigma \sim D}[q_\sigma^t(x) = b]$, we need to prove that there exists a value of $p$ such that: $p \cdot P_2^t + (1-p) \cdot P_6^t = (1-p) \cdot P_3^t + p \cdot P_5^t$.

Observe that for $t = 1$ and any $m \geq 7$ and for $t = 2$ and any $m \geq 8$ it holds that: $P_2^t = P_3^t = P_5^t = P_6^t$, for any value of $p$. If $m = 7$ and $t = 2$, then $P_2^t = P_3^t = P_5^t = 1$ and $P_6^t = 1/2$ and the above holds for $p = 1$. Now, we focus on the case of $m = 6$. In this case, if $t = 1$, $P_2^t = P_3^t = P_5^t = 1$ and $P_6^t = 0$, and the above is satisfied for $p = 1$. If $t = 2$, then $P_2^t = P_3^t = 1$, $P_5^t = 1/2$ and $P_6^t = 0$, and the above is satisfied for $p = \frac{2}{3}$. Thus, $D$ and $D^{a \leftrightarrow b}$ are $t$-indistinguishable.

We have shown above that in any case, there is a value of $p > 1/2$ such that $Pr_{\sigma \sim D}[q_\sigma^t(x) = a] = Pr_{\sigma \sim D}[q_\sigma^t(x) = b]$. This immediately implies that, $Pr_{\sigma \sim D}[a \succ_\sigma b] = p > Pr_{\sigma \sim D}[b \succ_\sigma a] = 1 - p$. Furthermore, for $x \in S_{-ab}$ it holds that $Pr_{\sigma \sim D}[a \succ_\sigma x] = \frac{m-3}{m-2} \cdot p + \frac{m-6}{m-2} \cdot (1-p)$ and $Pr_{\sigma \sim D}[b \succ_\sigma x] = \frac{m-5}{m-2} \cdot p + \frac{m-4}{m-2} \cdot (1-p)$. This is because with probability $p$, candidate $a$ is at position 2 and $b$ is at position 5, so a candidate $x$ below $a$ is sampled uniformly at random for the remaining $m - 3$ positions. Similarly, with probability $1 - p$, candidate $a$ is at position 6, so a candidate $x$ that is ranked below $x$, is sampled uniformly at random from the remaining $m - 6$ positions. Similarly, with probability $p$, candidate $b$ is at position 5, so $x$ is sampled uniformly at random for the remaining $m - 5$ positions. Furthermore, with probability $1 - p$, candidate $b$ is at position 3, with $a$ at position 6, so $x$ is sampled uniformly at random for the remaining $m - 4$ positions. Now, for the final step of the proof, we need the following inequality to be true:

$$\sum_{x \in S_{-ab}} Pr_{\sigma \sim D}[a \succ_\sigma x] > \sum_{x \in S_{-ab}} Pr_{\sigma \sim D}[b \succ_\sigma x]$$

$$\implies \sum_{x \in S_{-ab}} \frac{m-3}{m-2} \cdot p + \frac{m-6}{m-2} \cdot (1-p) > \sum_{x \in S_{-ab}} \frac{m-5}{m-2} \cdot p + \frac{m-4}{m-2} \cdot (1-p)$$

$$\implies (m-3) \cdot p + (m-6) \cdot (1-p) > (m-5) \cdot p + (m-4) \cdot (1-p)$$

$$\implies 2 \cdot p > 2 \cdot (1-p)$$

$$\implies p > {}^1\!/_2$$

As we have already shown, in any case, $p > {}^1\!/_2$. This combined with the fact that $Pr_{\sigma \sim D}[a \succ_\sigma b] > Pr_{\sigma \sim D}[b \succ_\sigma a]$, yields $\sum_{x \in M \setminus \{a\}} Pr_{\sigma \sim D}[a \succ_\sigma x] > \sum_{x \in M \setminus \{b\}} Pr_{\sigma \sim D}[b \succ_\sigma x]$, which concludes the proof of the case.

**Case III:** $t \in \{m-1, m-2, m-3\}$. Let $i = max(2, m-t)$. We set preference profile $D = \hat{D}_i$, where $\hat{D}_i$ is the preference profile from Lemma C.1. Hence for $p = \frac{i}{t+i} < {}^1\!/_2$, which holds for $t \geq m-3$ and $m > 6$, $D$ and $D^{a \leftrightarrow b}$ are $t$-indistinguishable. Observe that $Pr_{\sigma \sim D}[b \succ_\sigma a] = 1 - p$ and $Pr_{\sigma \sim D}[a \succ_\sigma b] = p$. Moreover, for each $x \in S_{-ab}$, $Pr_{\sigma \sim D}[a \succ_\sigma x] = p \cdot \frac{m-i-1}{m-2}$, as, with probability $p$, $a$ appears at position $i$, $b$ appears at position $i+1$, and then all the remaining candidates occupy the remaining $m-i-1$ positions uniformly at random. Furthermore, with probability $1-p$, candidate $a$ appears at position $m$. Similarly, $Pr_{\sigma \sim D}[b \succ_\sigma x] = p \cdot \frac{m-i-1}{m-2}$. Thus, we get that

$$\sum_{x \in M \setminus \{a\}} Pr_{\sigma \sim D}[a \succ_\sigma x] = \sum_{x \in S_{-ab}} Pr_{\sigma \sim D}[a \succ_\sigma x] + Pr_{\sigma \sim D}[a \succ_\sigma b] = \sum_{x \in S_{-ab}} \frac{m-i-1}{m-2} \cdot p + p,$$

and

$$\sum_{x \in M \setminus \{b\}} Pr_{\sigma \sim D}[b \succ_\sigma x] = \sum_{x \in S_{-ab}} Pr_{\sigma \sim D}[b \succ_\sigma x] + Pr_{\sigma \sim D}[b \succ_\sigma a] = \sum_{x \in M \setminus \{a,b\}} \frac{m-i-1}{m-2} \cdot p + 1 - p.$$

Now, since $p < {}^1\!/_2$, it follows that $\sum_{x \in M \setminus \{a\}} Pr_{\sigma \sim D}[b \succ_\sigma x] > \sum_{x \in M \setminus \{b\}} Pr_{\sigma \sim D}[a \succ_\sigma x]$ and this concludes the proof of the case.

$\square$

## G. Proof of Lemma 5.4

*Proof.* We start by considering the preference profile $D_{i,m,i+1}$ from Definition 3.1 for any $i \in \{2, \ldots, m-t-2\}$. From Lemma 3.2, we get that $D_{i,m,i+1}$ and $D_{i,m,i+1}^{a \leftrightarrow b}$ are $t$-indistinguishable when $p = P_{i+1}^t/(P_i^t + P_{i+1}^t)$. Now notice that

$$\sum_{x \in M \setminus \{a\}} Pr_{\sigma \sim D_{i,m,i+1}}[a \succ_\sigma x] = \sum_{x \in M \setminus \{b,a\}} \sum_{r=i}^{m-2} Pr[\tau^{S_{-ab}}(r) = x] \cdot p + Pr_{\sigma \sim D_{i,m,i+1}}[a \succ_\sigma b]$$

$$= \sum_{x \in M \setminus \{b,a\}} \frac{m-2-i+1}{m-2} \cdot p + p = (m-i) \cdot p,$$

and

$$\sum_{x \in M \setminus \{b\}} Pr_{\sigma \sim D_{i,m,i+1}}[b \succ_\sigma x] = \sum_{x \in M \setminus \{b,a\}} \sum_{r=i+1}^{m-2} Pr[\tau^{S_{-ab}}(r) = x] \cdot (1-p) + Pr_{\sigma \sim D_{i,m,i+1}}[b \succ_\sigma a]$$

$$= \sum_{x \in M \setminus \{b,a\}} \frac{m-2-(i+1)+1}{m-2} \cdot (1-p) + (1-p) = (m-i-1) \cdot p.$$

If for some $i \in \{2, \ldots, m-t-2\}$, we have that $(m-i) \cdot p \neq (m-i-1) \cdot (1-p)$, since $D_{i,m,i+1}$ and $D_{i,m,i+1}^{a \leftrightarrow b}$ are $t$-indistinguishable, by utilizing Lemma 5.1, we can construct a family of preference profiles $\{D^c\}_{c \in M}$ with $m$ distinct Condorcet winners that are $t$-indistinguishable. Then, by applying Lemma 3.3, we get the desired result.

Now suppose that $(m-i) \cdot p = (m-i-1) \cdot (1-p)$ for all $i \in \{2, \ldots, m-t-2\}$. Then, since $p = P_{i+1}^t/(P_i^t + P_{i+1}^t)$, we conclude that

$$\frac{P_i^t}{P_{i+1}^t} = \frac{m-i}{m-i-1}, \qquad \forall i = \{2, \ldots, m-t-2\} \tag{8}$$

Next, we consider the preference profile $D_{2,m-i,i+2}$ for any $i \in \{1, \ldots, t+1\}$. Since $t \leq m/2 - 2$, , from Lemma 3.2, we get that $D_{2,m-i,i+2}$ and $D_{2,m-i,i+2}^{a \leftrightarrow b}$ are $t$-indistinguishable when $p = P_{i+2}^t/(P_2^t - P_{m-i}^t + P_{i+2}^t)$. Next, we see that

$$\sum_{x \in M \setminus \{a\}} Pr_{\sigma \sim D_{2,m-i,i+2}}[a \succ_\sigma x] = \sum_{x \in M \setminus \{b,a\}} \sum_{r=2}^{m-2} Pr[\tau^{S_{-ab}}(r) = x] \cdot p + Pr_{\sigma \sim D_{2,m-i,i+2}}[a \succ_\sigma b]$$

$$= \sum_{x \in M \setminus \{b,a\}} \frac{m-2-2+1}{m-2} \cdot p + p = (m-2) \cdot p,$$

and

$$\sum_{x \in M \setminus \{b\}} Pr_{\sigma \sim D_{2,m-i,i+2}}[b \succ_\sigma x]$$

$$= \sum_{x \in M \setminus \{b,a\}} \sum_{r=m-i-1}^{m-2} Pr[\tau^{S_{-ab}}(r) = x] \cdot p$$

$$+ \sum_{x \in M \setminus \{b,a\}} \sum_{r=i+2}^{m-2} Pr[\tau^{S_{-ab}}(r) = x] \cdot (1-p) + Pr_{\sigma \sim D_{2,m-i,i+2}}[b \succ_\sigma a]$$

$$= \sum_{x \in M \setminus \{b,a\}} \frac{m-2-(m-i-1)+1}{m-2} \cdot p + \sum_{x \in M \setminus \{b,a\}} \frac{m-2-(i+2)+1}{m-2} \cdot (1-p) + (1-p)$$

$$= i \cdot p + (m-i-2) \cdot (1-p).$$

As before, if for some $i \in \{1, \ldots, t+1\}$, we have that $(m-2) \cdot p \neq i \cdot p + (m-i-2) \cdot (1-p)$, by utilizing Lemma 5.1, we can construct a family of preference profiles $\{D^c\}_{c \in M}$ with $m$ distinct Condorcet winners that are $t$-indistinguishable and by applying Lemma 3.3, we get the desired result.

Now suppose that $(m-2) \cdot p = i \cdot p + (m-i-2) \cdot (1-p)$ for all $i \in \{1, \ldots, t+1\}$. This means that $p = 1/2$, and since $p = P_{i+2}^t/(P_2^t - P_{m-i}^t + P_{i+2}^t)$, we get that $P_{m-i}^t = P_2^t - P_{i+2}$. Since $t \leq m/2 - 2$ and $P_i^t/P_{i+1}^t = (m-i)/(m-i-1)$ for all

$i \in \{2, \ldots, m - t - 2\}$, we conclude that

$$P_{m-i}^t = P_2^t - \frac{m - (i+2)}{m-2} P_2^t = \frac{i}{m-2} P_2^t, \qquad \forall i \in \{1, \ldots, t+1\} \tag{9}$$

Then, from Equation (8) and Equation (9), we get that the only case that we cannot construct two profiles with the desired property is when $P_i^t/P_{i+1}^t = (m-i)/(m-i-1)$ for all $i \in \{2, \ldots, m-2\}$, and this concludes the lemma. $\quad\square$

$\square$

## H. Condorcet winner and Deterministic Algorithms

First, we discuss in detail why the negative result of randomized rules cannot be extended in the case where $P_i^t/P_{i+1}^t = (m-i)/(m-i-1)$ for all $i \in \{2, \ldots, m-2\}$.

It is well-known that the Borda score of a candidate $c \in M$ is given by $\sum_{x \in M \setminus \{c\}} \Pr_{\sigma \sim D}[c \succ_\sigma x]$ (Brandt et al., 2016). A candidate $c$ is the Condorcet winner if $\Pr_{\sigma \sim D}[c \succ_\sigma x] > \Pr_{\sigma \sim D}[x \succ_\sigma c]$ for all $x \in M \setminus \{c\}$. This implies that when $c$ is a Condorcet winner, then $\Pr_{\sigma \sim D}[c \succ_\sigma x] > 1/2$ for all $x \in M \setminus \{c\}$, and therefore, the Borda score of $c$ must be strictly greater than $\frac{m-1}{2}$.

Now, assume for contradiction that we have a set of $m$ preference profiles $\{D^c\}_{c \in M}$ that are $t$-indistinguishable from one another, and suppose $c$ is the Condorcet winner in $D^c$. This means the score of $c$ in $D^c$ must be strictly greater than $\frac{m-1}{2}$. When $\vec{s}_t^* = (P_1^t, P_2^t, \ldots, P_m^t)$ where $P_i^t/P_{i+1}^t = (m-i)/(m-i-1)$ for all $i \in \{2, \ldots, m-2\}$, then we have that $\vec{s}_{Borda} \in span(\vec{s}_{plu}, \vec{s}_t^*)$. Therefore, from Theorem 4.4 we get that we can learn the Borda score of all the candidates. With similar arguments, as in the proof of Lemma 4.3, we can get that

$$sc_{\vec{s}_{Borda}}(a, D) = \sum_{i=1}^m P_i^t \cdot \Pr_{\sigma \sim D}[\sigma^{-1}(a) = i] + (m - 1 - P_1^t) \cdot \Pr_{\sigma \sim D}[\sigma^{-1}(a) = 1]$$

$$= \sum_{b \in M \setminus \{a\}} \Pr_{\sigma \sim D}[q_\sigma^t(b) = a] + (m - 1 - P_1^t) \cdot \Pr_{\sigma \sim D}[q_\sigma^t(a) = a].$$

From the definition of indistinguishability, we have that $\Pr_{\sigma \sim D^c}[q_\sigma^t(b) = a] = \Pr_{\sigma \sim D^{c'}}[q_\sigma^t(b) = a]$ for any $a, b, c, c' \in M$ and therefore from the above equality, we get that that each candidate $c$ must have the same Borda score across all $m$ preference profiles. Consequently, in any preference profile, the total sum of scores for all candidates would be strictly larger than $m \cdot \frac{m-1}{2}$. However, this is a contradiction because the sum of Borda scores of all candidates in any preference profile is exactly equal to $m \cdot \frac{m-1}{2}$. Therefore, such a family of preference profiles cannot exist for this case.

Next, we show that for $t \leq m - 4$, no deterministic rule can identify the Condorcet winner under *any* $t$-improvement feedback queries.

**Theorem H.1.** *For any $m \geq 6$ and any $t \leq m - 4$, no algorithm with access to $t$-improvement feedback queries can identify the unique Condorcet winner.*

*Proof.* Consider the following preference profile $D$:

- With probability $\frac{1}{3}$, we select $D_{2,m,3}$ from Definition 3.1.

- With probability $\frac{1}{3}$, we have $a \succ b \succ \tau^{S-ab}$, where $\tau^{S-ab}$ is a uniformly random ranking over all remaining candidates.

- With probability $\frac{1}{3}$, we have $b \succ a \succ \tau^{S-ab}$, where $\tau^{S-ab}$ is a uniformly random ranking over all remaining candidates.

Since $t \leq m - 4$, by Lemma 3.2 and the construction of $D$, we have that $D$ and $D^{a \leftrightarrow b}$ are $t$-indistinguishable when $p = \frac{P_3^t}{P_2^t + P_3^t}$ in the construction of $D_{2,m,3}$. Note that for each $x \in S_{-ab}$, we have $\Pr_{\sigma \sim D}[a \succ_\sigma x] > \frac{2}{3}$ and $\Pr_{\sigma \sim D}[b \succ_\sigma x] > \frac{2}{3}$.

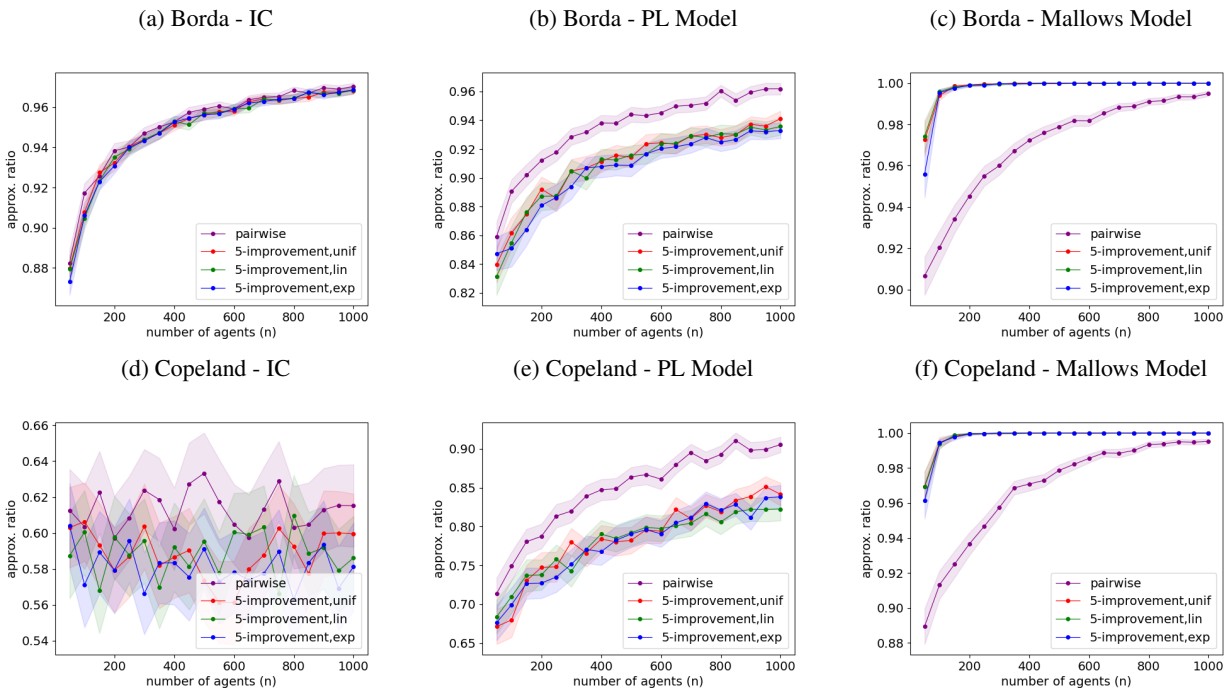

Figure 2: Different $t$-improvement feedback distributions

If $p \neq \frac{1}{2}$, then $\Pr_{\sigma \sim D}[a \succ_\sigma b] \neq \Pr_{\sigma \sim D}[b \succ_\sigma a]$, implying that one of $a$ or $b$ is the Condorcet winner. However, since $D$ and $D^{a \leftrightarrow b}$ are $t$-indistinguishable, no deterministic algorithm can always output the Condorcet winner.

On the other hand, if $p = \frac{1}{2}$, we have $\Pr_{\sigma \sim D}[a \succ_\sigma b] = \Pr_{\sigma \sim D}[b \succ_\sigma a]$. However, $\Pr_{\sigma \sim D}[a \succ_\sigma x] > \Pr_{\sigma \sim D}[b \succ_\sigma x]$ for all $x \in S_{-ab}$, because, in $D_{2,m,3}$, with probability $1/2 \cdot 1/3 = 1/6$, $a$ appears in the second position and $b$ in the last position, while with probability also $1/6$, $b$ appears in the third position and $a$ again in the last position, with all other candidates arranged uniformly at random. Hence there is one position (namely, position 3) where $x$ can be below $a$ but not $b$. In this case, by utilizing Lemma 5.1 and Lemma 3.3, we can show that even a randomized algorithm cannot find the Condorcet winner with probability greater than $1/m$. Therefore, no deterministic rule can always find the Condorcet winner, and the theorem follows. □

## I. More Experiments

In Figure 2, we present the approximation ratio of the Borda and Copeland scores across the three different ranking distributions and under the three different $t$-improvement feedback distributions. We observe that in all cases, the approximation ratio is similar.

In Figure 3, we illustrate the approximation ratio of the Borda and Copeland scores across the three different ranking distributions for varying values of $t$ and $n = 500$. Once again, we observe that the results do not significantly change for different values of $t$. However, in some cases, we observe that the approximation ratio improves as $t$ increases.

For our experiments, we used the Python package pref-voting.

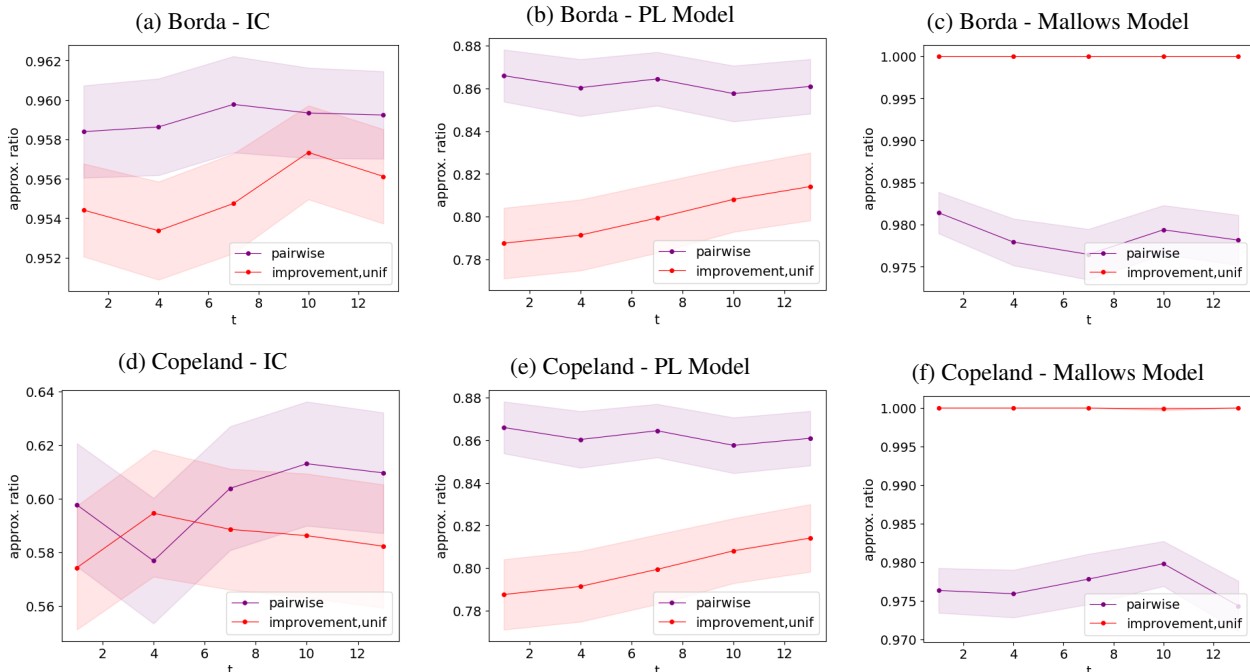

Figure 3: Varying values of $t$

