# OpenReview forum: "Computing Voting Rules with Improvement Feedback"
_ICML.cc/2025/Conference — ICML 2025 poster_

### Official Review · Reviewer_5mjJ · 2025-02-27

**Overall Recommendation:** 3

**Summary:**

This paper investigates the feasibility of computing voting rules using improvement feedback, a type of iterative preference refinement distinct from pairwise comparisons. It characterizes the positional scoring rules that can be learned under improvement feedback, demonstrating that while plurality can be determined, many other rules face strong impossibility results. The study also establishes that Condorcet-consistent rules cannot be computed using improvement feedback. Theoretical findings are supported by experimental analyses, showing practical implications of the proposed method.

**Claims And Evidence:**

- The authors claim that improvement feedback enables learning the plurality rule but fails for many other scoring rules.
- The paper provides theoretical proofs demonstrating that improvement feedback does not suffice to compute Condorcet-consistent rules.
- Empirical simulations support the theoretical findings, comparing improvement feedback and pairwise comparisons in approximating voting rule outcomes.

**Essential References Not Discussed:**

.

**Experimental Designs Or Analyses:**

- Simulations are conducted using synthetic ranking data generated from IC, Mallows, and Plackett-Luce models.
- Different t-improvement feedback distributions (uniform, linear decay, and exponential decay) are tested.
- The performance of improvement feedback is compared against pairwise comparison feedback for Borda and Copeland rules.
- Results show that under certain preference distributions, improvement feedback performs comparably or even better than pairwise feedback in approximating voting rule outcomes.

**Methods And Evaluation Criteria:**

- Theoretical analysis using formal models of voting and preference aggregation.
- Characterization of learnability for different voting rules using improvement feedback.
- Empirical evaluation through simulations on different preference distributions, including Impartial Culture (IC), Mallows, and Plackett-Luce models.
- Approximation ratios are used to measure the effectiveness of improvement feedback compared to pairwise comparison feedback.

**Other Comments Or Suggestions:**

- The authors could explore hybrid feedback mechanisms combining improvement feedback with pairwise comparisons to mitigate the limitations identified.
- More discussion on the implications of these results for participatory decision-making systems could enhance the broader impact of the work.
- It would be helpful to consider how strategic behavior by voters might impact the effectiveness of improvement feedback.

**Other Strengths And Weaknesses:**

### Strengths
- Provides a rigorous theoretical foundation for analyzing improvement feedback in voting.
- Well-structured proof techniques and clear characterization of learnable voting rules.
- Complementary empirical analysis enhances practical relevance.

### Weaknesses
- Limited discussion of how improvement feedback can be practically elicited in real-world applications.
- The focus on worst-case impossibility results may not fully capture practical performance in realistic scenarios.
- The paper assumes idealized access to feedback distributions, which may not hold in practical settings.

**Questions For Authors:**

Authors consider multiple answer setting, and in that case Condorset winner cannot be determined. However, getting multiple preference answers is costly and difficult and can be decomposed as getting pairwise comparisons n times. Why should we solve the problem which happens only under multiple answers?

**Relation To Broader Scientific Literature:**

- Extends prior work on voting rule computation under limited feedback (e.g., Halpern et al. 2024).
- Builds on research in coactive learning and interactive preference learning by examining feedback mechanisms for collective decision-making.
- Contributes to computational social choice by analyzing preference aggregation under practical constraints.

**Theoretical Claims:**

- Improvement feedback can determine the plurality winner but fails for most other positional scoring rules.
- No algorithm, deterministic or randomized, can identify a Condorcet winner using improvement feedback with a probability greater than $\frac{1}{m}$.
- The impossibility results hold under general improvement feedback distributions, except for a specific case where $P^t_i/P^t_{i+1}$ follows a defined ratio.

---

> ### Author Rebuttal · Authors · 2025-03-31
>
> We thank the reviewer.
>
> >Authors consider multiple answer setting, and in that case Condorset winner cannot be determined. However, getting multiple preference answers is costly and difficult and can be decomposed as getting pairwise comparisons n times. Why should we solve the problem which happens only under multiple answers?
>
> We would like to clarify that in the $t$-improvement model, each query returns only a single candidate—namely, a candidate from the $t$-above neighborhood of the queried option. The model does not assume multiple feedback points per query (lines 60-62).
>
> The impossibility results we derive are not due to assuming "multiple answers" but rather due to the restricted nature of the available feedback. While it's true that repeated pairwise comparisons could reconstruct full rankings, our focus is precisely on the types of settings where such fine-grained elicitation is infeasible (lines 38-40).
>
>
> >Limited discussion of how improvement feedback can be practically elicited in real-world applications.
>
> Improvement feedback models scenarios where users iteratively refine suggestions—e.g., editing LLM-generated text or adjusting robotic actions—by providing local, incremental improvements. This form of interaction is common in coactive learning, preference-based reinforcement learning, and broader human-in-the-loop decision-making systems. We will clarify this motivation even more in the introduction.
>
>
> >The focus on worst-case impossibility results may not fully capture practical performance in realistic scenarios.
>
> As is typical in theoretical work, our analysis focuses on worst-case guarantees to characterize the fundamental limitations of improvement feedback. However, we agree that worst-case behavior may not reflect typical performance in realistic settings. That is why we conducted experiments on more realistic distributions—including IC, Mallows, and Plackett-Luce—which go beyond the worst-case. These experiments show that improvement feedback can perform competitively with, and in some cases even outperform, pairwise comparisons in approximating voting rule outcomes. This contrast highlights the practical value of improvement feedback despite its theoretical limitations.
>
> >The paper assumes idealized access to feedback distributions, which may not hold in practical settings.
>
> Indeed, access to the exact feedback distribution is an idealized assumption, which **only strengthens our negative results**: if impossibility holds under full information, it certainly holds in more realistic settings with partial or noisy access (lines 80-84, 189-192). However, in practice, these distributions can be approximated via sufficiently many user queries, and one can formally justify this using standard concentration techniques.
>
> >The authors could explore hybrid feedback mechanisms combining improvement feedback with pairwise comparisons to mitigate the limitations identified.
>
> We agree this is a compelling direction, as noted in the third sentence of our Discussion section. It is a question we have also given considerable thought. The main challenge is that the indistinguishable profiles constructed by Halpern et al. for pairwise comparisons are fundamentally different from those we construct for improvement feedback. Their profiles (respectively, ours) lead to impossibility under pairwise comparisons (respectively, improvement feedback), but do not remain indistinguishable when accessed via the other feedback model. As a result, making progress in this setting would require either (i) extending the impossibility results to hold even when both types of feedback are available, which would require new indistinguishable constructions, or (ii) showing that access to both feedback models enables algorithms to recover the correct winner, by designing techniques that exploit their complementary strengths. We leave this question to future work.
>
> >More discussion on the implications of these results for participatory decision-making systems could enhance the broader impact of the work.
>
> Our work highlights fundamental limitations in inferring collective decisions from restricted forms of feedback, which are common in real-world. Understanding which social choice rules are robust under such bounded input is crucial for designing systems that are both practical and representative. We will expand the discussion in the final version to better highlight these connections.
>
>
> >It would be helpful to consider how strategic behavior by voters might impact the effectiveness of improvement feedback.
>
> Our negative results already hold under the assumption that voters report their preferences truthfully. Introducing strategic behavior would  **weaken** these impossibility results, as it introduces an additional layer of uncertainty and misalignment between observed feedback and true preferences. However, we agree that understanding strategic behaviors in this setting is an interesting direction for future work.

---

### Official Review · Reviewer_dfHM · 2025-03-08

**Overall Recommendation:** 3

**Summary:**

This paper discusses the potential for any learning algorithm to identify a social choice winner f when sampling preferences from a distribution D. It specifically talks about t-improvement feedback, where D is hidden but can be sampled such that -- given a candidate $a$ in position $i$, one of the t candidates that are higher ranking that $i$ can be elicited at a time. Their main theorems distinguish certain sufficient conditions where an algorithm A cannot identify the winner f(.) when they have access to t-improvement feedback, in the worst-case, under position scoring rules and Condorcet-consistent rules.

*NOTE: Score changed 2->3 after reading authors' rebuttal.*

**Claims And Evidence:**

It could be made clearer how the failure of t-improvement feedback entails their claim that no algorithm can learn the correct output f(.) with high enough probability. The paper is confusing about what information algorithms A have access to, and specifically how they conduct t-improvement feedback. I would like to see this spelled out more explicitly.

**Essential References Not Discussed:**

In Related Work (Sec 1.2), talk about relationship of your work to the possible and necessary winner problems, as introduced by [1,2] (see also Sec 10.3.1 of (Brandt et al., 2016). I would like to see how your results compare to the impossibility results presented by this prior line of research. Also, how do your query complexity findings compare to other preference learning research, such as with pairwise comparisons in the Bradley-Terry model?

[1] Kathrin Konczak and Jérôme Lang. 2005. Voting procedures with incomplete preferences.
[2] Lirong Xia and Vincent Conitzer. 2011. Determining Possible and Necessary Winners under Common Voting Rules Given Partial Orders.

**Experimental Designs Or Analyses:**

Experiments look good. They support and provide complementary insight into the paper's main claims.

**Methods And Evaluation Criteria:**

Yes.

**Other Comments Or Suggestions:**

Minor:
- (Typo) In Model (Sec 2), you use $\mathcal{L}$ and $L$ interchangeably.
- In Model (Sec 2), you refer to "pi of D" as a sample from the distribution D subject to the permutation pi. Is this supposed to be a sample, or a modified distribution itself?
- (p4 LHS) Please define $q^t_\sigma$.
- Format of Lemma 3.1: take the definition of $D_{i,j,l}$ out of the lemma statement.
- (Typo) In Lemma 3.2: "...suppose there are is a family..."
- (Typo) In Lemma 4.1, you say "indistinguishable" instead of "t-indistinguishable."
- In main body, tell the reader that Lemma 4.1 is proved in Appendix B
- Proof of Lemma 4.2: is $\lambda$ defined as $s_i / P^t_i$ here? How is this related to $\vec{s}^*_t$?

**Other Strengths And Weaknesses:**

**Overall review:**

The paper does a good job posing an interesting question -- the learnability of outputs f given t-improvement feedback -- and taking steps to solve it under various social choice functions f. It is well-put into context of other work to learn social choice functions. It checks off the boxes I'd like to see out of this type of paper: pose interesting question, frame the model, non-trivial theoretical findings, experiments and discussion.

However, there are certain questions I still have:
- Certain aspects of the proofs are not clear
- How the inability to distinguish between two distributions leads to the claim that algorithms cannot learn f better than random
- What information algorithms A have access to when conducting their queries
- How these results compare to the theoretical query complexity of t-improvement feedback or pairwise feedback
- How these results compare to the NP-hardness of the possible/necessary winner problems

Also:
- The paper seems to significantly build on (Halpern et al., 2024), though doesn't go into too much detail on the work.
- What motivates the "t-improvement" concept? In (p2 LHS) you say t reflects the "limited cognitive and practical effort" of users, though this seems like a limited jumping-off point.

Therefore I would recommend the paper gets edited and resubmitted. If the authors can point to the aspects of this paper that address these issues, or answer them simply, I'd be willing to reconsider.

**Questions For Authors:**

NA

**Relation To Broader Scientific Literature:**

For the most part, yes. The authors discuss relevant literature in partial-order preference elicitation and query complexity. They don't go too much into depth in the literature about pairwise elicitation, such as papers discussing elicitation in the the Bradley-Terry model.

**Theoretical Claims:**

- In Lemma 3.1 why does the requirement on p need to hold? Doesn't "j-i > t" and "m-l > t" suggest that querying either a or b will (necessarily) never yield the other, since it's outside the scope of t items?
- Please clarify the second-to-last sentence in the proof of Lemma 4.2. This doesn't seem to follow from prior deductions. The second section of this proof appears quite round-a-bout and I'm not sure why it's there.
- In proof of Thm 4.4, why can we apply Lemma 3.2? Doesn't that require m distinct distributions, whereas the construction from Lemma 4.1 only guarantees some family of distributions? Also -- in the second to last sentence, why is the scoring rule in span(*,*)? Isn't this part discussing when the scoring rule isn't in span(*,*)?

---

> ### Author Rebuttal · Authors · 2025-03-31
>
> We thank the reviewer for their thoughtful and constructive feedback.
>
>
> >The paper is confusing about what information algorithms A has access to, and specifically how they conduct t-improvement feedback.
>
> As noted in your summary and in lines 60–62, under the $t$-improvement feedback model, when a candidate  $a$ is queried, the response is a random candidate drawn from the $t$-above neighborhood of $a$ in the voter's ranking. This candidate is sampled according to a fixed $t$-improvement feedback distribution, which defines the probabilities over the  $t$-above candidates (lines 63–64). Given a preference profile and a $t$-improvement feedback distribution, this sampling procedure induces a marginal probability of observing $b$ as feedback when querying  $a$.
>
> While in practice, such marginals can be estimated by sufficiently many samples, here we make the strong assumption of full information: the algorithm has exact access to the probabilities of seeing $b$ when querying $a$, for every pair $a,b\in M$ (lines 74–81). This idealized access **strictly strengthens our impossibility results**--if no algorithm can succeed finding the winner with full information, it cannot succeed with less (see lines 82–84, 189-192).
>
> >How these results compare to the theoretical query complexity of t-improvement feedback or pairwise feedback.
>
> Our results do not address query complexity (lines 104–108 rhs). As noted above, we assume idealized access to the exact marginal probabilities of receiving $b$ when querying $a$, thereby abstracting away sampling concerns. As such, our results are orthogonal to work in preference learning that focuses on sample efficiency (e.g., under the Bradley–Terry model).
>
> >How the inability to distinguish between two distributions leads to the claim that algorithms cannot learn f better than random.  proof of Thm 4.4, why can we apply Lemma 3.2?
>
> Indeed, Lemma 3.2 requires the existence of  $m$ preference profiles that are all $t$-indistinguishable from each other and each have a unique winner. Lemma 4.1 constructs exactly such a set of profiles, as stated explicitly. We suspect the confusion may stem from our use of the term “family of profiles”, which we use in line with standard terminology (see Lemmas 4.1–4.3 in Halpern et al.).
>
> >In the second-to-last sentence, why is the scoring rule in span(,)?
>
> This is a typo. The sentence should refer to scoring vectors  **not in** the span. We apologize for it.
>
> >In Lemma 3.1 why does the requirement on p need to hold? Doesn't "j-i > t" and "m-l > t" suggest that querying either a or b will (necessarily) never yield the other, since it's outside the scope of t items?
>
> Indeed, these conditions indicate that querying either $a$ or $b$ will never yield the other, so indeed by swapping them the probability of asking one and seeing the other remains the same  (equal to $0$). However, to ensure that the two profiles are indistinguishable, we must guarantee that for **every** $x,y \in M$ the probability of querying $x$ and observing $y$ is the same under both profiles (see definition in lines 195-200). The value of $p$ is chosen to maintain this property (see lines 550–582).
>
>
>
> >Please clarify the second-to-last sentence in the proof of Lemma 4.2.
>
> First part of the proof concerns positions $2$ to $ m-t-1$ (line 263). Second part of the proof concerns positions $m-t$ to $m-1$ (line 234 rhs).  We relate position $2$  to positions $m-t$ to $m-1$  in order to argue that the ratio $s_i/ P_i$ is the same (and equal to $\lambda$) for all positions from $2$ to $m-1$.
>
> >How these results compare to the NP-hardness of possible/necessary winner problems.
>
> That line of work  focuses on computational complexity of determining whether a candidate is a possible or necessary winner given partial orders. In contrast, we assume access to full statistical feedback and study whether any algorithm—regardless of computational complexity—can recover the winner. Our impossibility results are information-theoretic, not complexity-theoretic (see lines 104-106).
>
> >What motivates the “t-improvement” concept?
>
> The notion of $t$-improvement captures interactive settings where users refine suggestions through small, local adjustments—e.g., modifying LLM outputs or adjusting robot behavior—rather than providing full rankings. This interaction model appears in frameworks such as coactive learning and interactive preference learning (see lines 25–35 rhs and 110–114). The parameter $t$ captures the granularity of improvement.
>
> >The paper seems to significantly build on Halpern et al. (2024), though doesn't go into much detail on the work.
>
> While our work is conceptually related to Halpern et al, the technical contributions are distinct. The preference profiles we construct differ significanlty and are tailored to the $t$-improvement feedback model rather than pairwise comparisons. Where we do draw on Halpern et al.—e.g., in Lemma 3.2—we reprove the results in our setting for completeness.

---

> > ### Comment · Reviewer_dfHM · 2025-04-08
> >
> > Thank you for the comments. I have improved my score accordingly. I still would like to see this paper edited by incorporating reviewers' comments prior to publication.

---

> > > ### Author Response · Authors · 2025-04-08
> > >
> > > We sincerely thank the reviewer for taking the time to carefully read our rebuttal and revise their score accordingly. We promise to revise the paper by incorporating all reviewers' very valuable and constructive feedback.

---

### Official Review · Reviewer_5Qt1 · 2025-03-12

**Overall Recommendation:** 4

**Summary:**

The paper studies the concept of "improvement feedback" within computational social choice. Improvement feedback is a response given to a query of a specific voter asking a question along the lines of "which candidate do you prefer over x?" The main question of the paper is whether queries in the improvement feedback setting provide enough information to compute the winner of various voting rules.

The paper provides a few possibility results: It is possible to compute the plurality winner and any positional scoring rule between plurality and a specific scoring rule determined by the setting itself. The paper shows that every other positional scoring rule cannot be exactly computed using this framework. Similarly, it is shown to be impossible to compute the Condorcet winner.

Experiments show that while it is impossible to compute the winner of many voting rules in theory, it is often possible to do so in practice. Using improvement feedback to estimate the winner of the Borda and Copeland rules shows that across three different preferences profiles, it is frequently possible to achieve a high accuracy at estimating these rules. Improvement feedback is compared with the accuracy of estimating these rules using pairwise feedback and shown to be more accurate in some settings/less accurate than others.

**Claims And Evidence:**

The claims made in the paper are supported fairly well.

**Essential References Not Discussed:**

N/A

**Experimental Designs Or Analyses:**

The experiments are quite reasonable and appropriate for the question at hand. Slightly more explanation would be useful. Specifically, an explanation of the approximation ratio -- I think I know how this is defined here but I can imagine a few semi-reasonable definitions.

**Methods And Evaluation Criteria:**

The paper makes theoretical claims and supports them appropriately. Experiments add a very interesting new depth to the theoretical results.

**Other Comments Or Suggestions:**

I noticed several minor typos. I would encourage re-reading to catch any that I did not notice.

Line 128, Column 1: You use $\mathcal{L}$ here and $L$ elsewhere in the paper.

L194C2: "there are is a family..."

L205C2: "any of this preference profiles"

L309C2 (in the equation): Should there be a $D$ subscript for the probabilities?

L334/335C2: The sentence "For both rules suffices..." needs rearranging.

L351C2: "is assigned a r weight"

**Other Strengths And Weaknesses:**

Overall I find the main idea of the paper to be quite reasonable and generally like the paper. This fits well into existing and ongoing work into learning about partial preferences.

There is some room for improvement in presentation which I would encourage should the opportunity arise. In particular, I would suggest ensuring that the introduction is primarily focused on building intuition around the paper. This is largely the case but the section could be further simplified. A stronger connection to possible applications would also be useful, if you do see such a connection. Including slightly more natural language/intuition-building throughout the paper would also improve the readability of the work.

I very much like that you have included some experiments. That it is usually possible to effectively learn Borda/Copeland despite not being possible to guarantee highlights, for me, the importance of doing experiments to actually understand the behaviour of a setting. Additional details on the setup are important, however. Including them in the appendix would be fine. Specifically, I would hope for more detail on the distributions (is Mallow's using Boehmer's re-parameterization?) and the winner determination.

I am interested in winner determination because I find the poor accuracy at learning the Copeland winner to be quite surprising. Is the algorithm learning a single weighted majority graph then using it to compute both the Borda and Copeland winners (also, is weighted majority graph defined in the paper?)? Then what is actually being learned from the impartial culture profile when Copeland is so poorly estimated? Perhaps I am missing it but some discussion about the accuracy of Copeland under IC seems quite important.

Additionally, if you are learning a weighted majority graph using the pref-voting package it should be quite trivial to compute the winner of all C1/C2 rules in the package. While not the main focus of the paper, doing this could be quite interesting (especially given that you already know there is at least one rule, Copeland, which exhibits surprising behaviour).

**Questions For Authors:**

Do you have any explanation for the relatively poor behaviour of Copeland in the experiments? I found this quite surprising -- is it also surprising to you?


(A response is unlikely to change my evaluation but I am quite curious about this.)

**Relation To Broader Scientific Literature:**

The paper builds upon one recent work studying comparison feedback and, less directly, several other papers considering partial information settings and t-wise queries.

This is a very natural extension of existing research and could conceivably relate to some applications, such as in RLHF.

**Theoretical Claims:**

I lightly reviewed the proofs included in the main text of the paper.

---

> ### Author Rebuttal · Authors · 2025-04-01
>
> We thank the reviewer for their thoughtful and constructive feedback.
>
> > Slightly more explanation would be useful. Specifically, an explanation of the approximation ratio -- I think I know how this is defined here but I can imagine a few semi-reasonable definitions.
>
> For a fixed number of agents (n) and a single trial (out of 500), the approximation ratio is defined as follows: we identify the winner using the partial feedback and compute their true score under the full preference profile. This is then divided by the true score of the actual winner, i.e., the candidate who would have been selected using full information. We average this approximation ratio over all 500 trials and report both the mean and standard deviation. We will clarify this point in the revised version.
>
>
> >Additional details on the setup are important, however. Including them in the appendix would be fine. Specifically, I would hope for more detail on the distributions (is Mallow's using Boehmer's re-parameterization?) and the winner determination.
>
>
> We do not use Boehmer’s re-parameterization, as the number of candidates remains fixed throughout our experiments. We use the classical definition of the Mallows model, where, given a central ranking $\sigma^*$ and a parameter $\phi \in (0, 1]$, the probability of observing a ranking $\sigma$ is defined as:
>
> $\frac{\phi^{d(\sigma, \sigma^*)}}{\sum_{\sigma' \in \mathcal{L}(A)} \phi^{d(\sigma', \sigma^*)}}$.
>
> where $d(\sigma, \sigma^*)$ denotes the Kendall tau distance between $\sigma$ and $\sigma^*$.
>
> For winner determination, we construct a weighted majority graph based on the available feedback. Specifically, under pairwise feedback, each voter is asked to compare a randomly selected pair of candidates and under $t$-improvement feedback, each voter is queried with a randomly selected candidate and returns one candidate from the $t$-above neighborhood of the queried candidate, according to the $t$-improvement feedback distribution. These responses collectively define a weighted majority graph, which is then used to determine the winner under each feedback type.
>
> We will make sure to include more detailed descriptions of the setup, including formally defined the weighted majority graph.
>
> > Is the algorithm learning a single weighted majority graph then using it to compute both the Borda and Copeland winners (also, is weighted majority graph defined in the paper?)? Then what is actually being learned from the impartial culture profile when Copeland is so poorly estimated?
>
> Yes, the same graphs are used for both Borda and Copeland.
>
> We also found this behavior surprising. But our conclusion is that it stems from the statistical nature of pairwise comparisons in random preference profiles. Under the IC model, most pairwise majority margins are extremely narrow—typically close to 50-50.  When we sample comparisons uniformly at random, there is a significant chance that sampling noise flips the majority outcome of a pair. For example, if candidate $a$ defeats  $b$ with 51% of the vote, the number of observed comparisons  may still favor $b$ in the sample. In such cases,  $a$ is assigned $0$ points instead of $1$ in the Copeland score. This effect cannot be avoided even when  the number of queries grows.
>
> Borda, on the other hand, is more robust in this setting. Even if the sampled pairwise data slightly misrepresents the majority—for example, favoring $b$ over $a$ 51% to 49%—the inferred Borda score for $a$ still reflects this proportionally. Since the Borda score can be interpreted as the sum of the estimated probabilities of a candidate defeating each other candidate (see line 1230), $a$ would still receive 0.49 points rather than being penalized with a full loss. This smoothness makes Borda less sensitive to small fluctuations.
>
> >Additionally, if you are learning a weighted majority graph using the pref-voting package it should be quite trivial to compute the winner of all C1/C2 rules in the package.
>
> We considered computing additional C1/C2 rules beyond Borda and Copeland. However, we ran into limitations stemming from the nature of the rules themselves. In particular, to make our definition of approximation ratio meaningful, we must restrict attention to rules that assign scores to candidates—otherwise, it's unclear how to compare the true and estimated winners. This requirement excludes many rules in the pref-voting library, such as Kemeny, Slater, and the Uncovered Set.
>
> We also considered the minimax rule, which does assign scores. However, we encountered technical issues that made it difficult to include. In particular, the true winner often has a minimax score of zero, resulting in undefined 0/0 approximation ratios. Additionally, minimax scores are typically nonpositive, with the winner being the candidate with the least negative value—this makes interpreting approximation ratios less meaningful. For these reasons, we ultimately decided not to include minimax in our evaluation.

---

### Decision · Program_Chairs · 2025-05-01

**Decision:**

Accept (poster)

**Comment:**

Reviewers felt that the problem investigated in this paper to be interesting, the theoretical results being solid, and the experiments being reasonably convincing. Some questions were clarified during the rebuttal and the overall sentiment became positive. There is still room for improvement in presentation as multiple reviewers noticed, but overall this is a nice addition to ICML and a nice contribution to computational social choice.